



# Measurement report: Long-term changes in black carbon and aerosol optical properties from 2012 to 2020 in Beijing, China

Jiaxing Sun[1,2], Zhe Wang[1], Wei Zhou[1], Conghui Xie[1,2, a], Cheng Wu[3], Chun Chen[1,2], Tingting Han[1, b],
Qingqing Wang[1], Zhijie Li[1,2], Jie Li[1], Pingqing Fu[4], Zifa Wang[1,2,5], and Yele Sun[1,2,5] *

[1]State Key Laboratory of Atmospheric Boundary Layer Physics and Atmospheric Chemistry, Institute of Atmospheric Physics, Chinese Academy of Sciences, Beijing 100029, China

[2]College of Earth and Planetary Sciences, University of Chinese Academy of Sciences, Beijing 100049, China

[3] Institute of Mass Spectrometer and Atmospheric Environment, Jinan University, Guangzhou 510632, China

[4]Institute of Surface-Earth System Science, Tianjin University, Tianjin 300072, China

[5]Center for Excellence in Regional Atmospheric Environment, Institute of Urban Environment, Chinese Academy of Sciences, Xiamen 361021, China

[a]now at: State Key Joint Laboratory of Environmental Simulation and Pollution Control, College of Environmental
Sciences and Engineering, Peking University, Beijing, 100871, China

[b]now at: Environmental Meteorology Forecast Center of Beijing-Tianjin-Hebei, Beijing 100089, China

* *Correspondence to:* Yele Sun (sunyele@mail.iap.ac.cn)

**Abstract.** Atmospheric aerosols play an important role in radiation balance of the earth-atmosphere system. However, our
knowledge of the long-term changes in black carbon (BC) and aerosol optical properties in China are very limited. Here we analyze the nine-year measurements of BC and aerosol optical properties from 2012 to 2020 in Beijing, China. Our results showed large reductions in eBC by 67 % from $5.54 \pm 5.25$ μg m$^{-3}$ in 2012 to $1.80 \pm 1.54$ μg m$^{-3}$ in 2020, and 47 % decreases in light extinction coefficient ($b_{ext}$, $\lambda = 630$ nm) of fine particles due to clean air action plan since 2013. The seasonal and diurnal variations of eBC illustrated the most significant reductions in the fall and night time, respectively.
ΔeBC/ΔCO also showed an annual decrease from ~ 6 to 4 ng m$^{-3}$ ppbv$^{-1}$ and presented strong seasonal variations with high values in spring and fall, indicating that primary emissions in Beijing have changed significantly. As a response to clean air action, single scattering albedo (SSA) showed a considerable increase from $0.79 \pm 0.11$ to $0.88 \pm 0.06$, and mass extinction efficiency (MEE) increased from 3.2 to 3.8 m$^2$ g$^{-1}$. These results highlight an increasing importance of scattering aerosols in radiative forcing, and a future challenge in visibility improvement due to enhanced MEE. Brown carbon (BrC)
showed similar changes and seasonal variations to eBC during 2018 – 2020. However, we found a large increase of secondary BrC in the total BrC in most seasons, particularly in summer with the contribution up to 50 %, demonstrating an enhanced role of secondary formation in BrC in recent years. The long-term changes in eBC and BrC have also affected





the radiative forcing effect. The direct radiative forcing ($\Delta F_R$) of BC decreased by 64 % from +3.00 W m$^{-2}$ in 2012 to +1.09 W m$^{-2}$ in 2020, and that of BrC decreased from +0.30 to +0.17 W m$^{-2}$ during 2018-2020. Such changes might have

important implications in affecting aerosol and boundary-layer interactions and the future air quality improvement.

## 1 Introduction

Atmospheric aerosols play an important role in radiative balance of the earth-atmosphere system by directly scattering and absorbing solar radiation or indirectly changing cloud reflectivity and precipitation processes (Bond and Bergstrom, 2006; Rosenfeld, 2000). Accurate assessment of radiative forcing caused by aerosol is still a challenge (IPCC, 2013) due to the

uncertainties in estimating scattering coefficient ($b_{sca}$), absorption coefficient ($b_{abs}$), and single-scattering albedo (SSA). Especially, SSA is a key factor determining whether aerosols exert a warming or cooling effect. Previous studies found that the increase of SSA from 0.8 to 0.9 can often shift the radiative forcing from positive to negative (Hansen et al., 1997; Lee et al., 2007). These key parameters of aerosol optical properties are closely related to the size distribution, mass concentration and composition of aerosols which have been extensively studied in previous studies (Han et al., 2015; Paola

Massoli et al., 2015; Xie et al., 2019; Zhai et al., 2017). Generally, sulfate, nitrate and a majority of organics predominantly scatter light and exert negative forcing effect (Han et al., 2015; Haywood and Boucher, 2000). Differently, black carbon (BC) and brown carbon (BrC) are the major absorption aerosols which warm atmosphere and present positive forcing effect (Bond et al., 2013). BC is mainly generated from the incomplete combustion of fossil fuels and bio-fuels which shows absorptions at all wavelengths and produces strong radiative forcing effect (Jacobson, 2001; Bond et al., 2013; IPCC,

2013). Depending on the measurement methods, BC is also called elemental carbon (EC), refractory black carbon (rBC) and equivalent black carbon (eBC) which is derived from converting light absorption coefficient into mass concentration with a suitable mass absorption efficient (MAE) (Petzold et al., 2013). As a part of organic carbon, the absorbing ability of BrC depends strongly on wavelength (Andreae and Gelencsér, 2006; Cappa et al., 2019), and generally accounts for 20 ~ 40 % to the total absorption of carbonaceous aerosol over a global scale (Saleh et al., 2013; Jo et al., 2016; Park et al.,

2010; Wang et al., 2019a). Besides the similar primary sources with BC (i.e., biomass burning, coal combustion and vehicle exhaust), BrC can also be produced from multiphase reactions like photochemical or aqueous-phase oxidation of volatile organic compounds (Laskin et al., 2015).

High concentrations of absorbing and scattering aerosols also cause air pollution and effect human's health (Oberdörster and Yu, 1990). BC is a particular pollutant which could affect the development of boundary layer by changing atmospheric

heating rate and then aggravate air pollution (Ding et al., 2016). Severe air pollution has been a wide environmental concern in China during the last decade (Zhou et al., 2020; An et al., 2019) . The previous studies showed that China and India contributed most to the BC emissions in Asia, accounting for 25 ~ 35 % 2010 (Li et al., 2017; Wei et al., 2020; Bond et al., 2013; Ramanathan and Carmichael, 2008). In China, BC emissions were estimated to be approximately 2534 Gg in 2014, twice as much as that in 1960 (Hoesly et al., 2018). Until 2017, the residential and industry contributed more than 83 % of

Chinese BC emissions (Wang et al., 2012). Focusing on BC sources and emissions changes in China has outstanding implications for global climate change.





Beijing as one of the largest megacities in the world has been a great success in decreasing $PM_{2.5}$ during the last decade by implementing clean air action plan (Zhang et al., 2019). Many previous studies focused on the changes in aerosol chemical components and the influences of emissions and meteorological conditions (Lei et al., 2020; Sun et al., 2020b; Sun et al., 2018). The mass concentration, mixing state, optical property and coating chemical composition of BC in Beijing were also widely investigated in field campaigns in specific seasons (Din et al., 2019; Liu et al., 2017; Sun et al., 2021; Xie et al., 2020; Han et al., 2017). However, our understanding of the long-term changes of black carbon, aerosol optical properties and radiative effects as a consequence of clean air action are very limited.

In this study, we conducted nine-year measurements of eBC and light extinction coefficient ($\lambda = 630$ nm) by using Aethalometers along with cavity attenuated phase shift (CAPS) extinction monitor in Beijing. The long-term changes in eBC, $b_{ext}$, SSA and mass extinction efficient (MEE) are investigated, and their annual, seasonal and diurnal variations are elucidated. Moreover, we illustrate the changes in BrC absorption and absorption Ångström exponent (AAE) by using three-year measurements from 2018 to 2020. Particularly, the contributions of primary emissions and secondary formation to BrC absorptions are quantified and their changes during the past three years are demonstrated. Finally, the impact of the changes in BC and BrC on direct radiative forcing is estimated and discussed.

## 2 Methods

### 2.1 Sampling sites and measurements

All optical measurements were conducted at the Institute of Atmospheric Physics (IAP), Chinese Academy of Sciences (39°58′28″N, 116°22′16″E) in Beijing More detail descriptions of this site were given in previous study (Sun et al., 2020a). Equivalent BC (eBC) were measured by a two wavelength (375 and 880 nm) Aethalometer (AE22) from August 2012 to December 2014 and a seven-wavelength (370, 470, 520, 590, 660, 880, and 950 nm) Aethalometer (AE33) from January 2015 to November 2020, along with the measurements of light extinction ($b_{ext}$, $\lambda = 630$ nm) of dry fine particles using a CAPS extinction monitor from 2012 to 2020. A more detailed instrument deployment is shown in Fig. S1. Note that the measurements of Aethalometers and CAPS from June 2013 to September 2014 and from September 2015 to August 2017 were not available. The AE22 and AE33 were operated at time resolutions of 10 min and 1 min, respectively, and the CAPS was operated at a time resolution of 1 s. Because the new version of AE33 using "dual-spot" technique can provide more reliable measurements by better correcting the filter-based loading effects (Drinovec et al., 2015), we further corrected the eBC measurements of AE22 according to parallel measurements (Han et al., 2015). The mass concentrations of $PM_{2.5}$ and CO were obtained from the air quality monitoring station at the Olympic Center (39°59′11″N, 116°23′58″E), which is approximately 4 km from our sampling site. The meteorological parameters of wind direction (WD) and wind speed (WS) were measured at the height of 102 m on the Beijing 325 m meteorological tower. In this study, four seasons are defined as: spring (March, April and May), summer (June, July and August), autumn (September, October and November), and winter (December, January and February in next year) in the following discussion.

### 2.2 Calculations of aerosol optical properties

Single scattering albedo (SSA, $\lambda = 630$ nm) of $PM_{2.5}$ can be calculated:





$$SSA = \frac{b_{ext} - b_{abs}}{b_{ext}} \tag{1}$$

$$b_{abs} = eBC \times MAE \tag{2}$$

where $b_{ext}$ is the light extinction coefficient at 630 nm. The mass concentration of eBC was converted to $b_{abs}$ at 630 nm using MAE of 7.9 in spring and summer and 7.4 in fall and winter, respectively (Han et al., 2017).

The mass extinction efficiency (MEE) of $PM_{2.5}$ was derived as the ratio of $b_{ext}$ to the mass concentration of $PM_{2.5}$,

$$MEE = \frac{b_{ext}}{PM_{2.5}} \tag{3}$$

The absorption Ångström exponent (AAE) can be determined using Eq. (4) (Moosmüller et al., 2011), and the $b_{abs,BC}$ at each wavelength was estimated assuming an AAE = 1 for pure BC (Bond and Bergstrom, 2006). After subtracting $b_{abs,\,BC}$ from the total absorption coefficient $b_{abs,\,total}$, the BrC absorption coefficient ($b_{abs,\,BrC}$) can be estimated with Eq. (5). In this

study, we calculated BrC absorption at 370 nm that referred to $b_{abs,\,BrC}$. Note that we may slightly overestimate the absorption of BrC due to the influence of dust though the MAE of dust was much lower than BC and BrC (Yang et al., 2008),.

$$\frac{b_{abs,\lambda1}}{b_{abs,\lambda2}} = \left(\frac{\lambda_1}{\lambda_2}\right)^{-AAE} \tag{4}$$

$$b_{abs,BrC} = b_{abs,total} - b_{abs,BC} \tag{5}$$

**2.3 Quantification of primary and secondary BrC absorption**

The absorption of BrC at 370 nm can be segregated into primary ($b_{abs,\,Primary\,BrC}$) and secondary ($b_{abs,\,Secondary\,BrC}$) using Eq. (6) assuming negligible contribution of dust.

$$b_{abs,\ Secondary\ BrC} = b_{abs,BrC} - (b_{abs,BrC}/b_{abs,BC})_{pri} \times b_{abs,BC} \tag{6}$$

where $b_{abs,\,BC}$ was the absorption at 880 nm, $(b_{abs,\,BrC}/b_{abs,\,BC})_{pri}$ is the ratio of primary BrC absorption to BC absorption.

Considering $b_{abs,\,BC}$ = [eBC]*MAE, we simplify the Eq. (6) to Eq. (7).

$$b_{abs,\ Secondary\ BrC} = b_{abs,BrC} - (b_{abs,BrC}/eBC)_{pri} \times eBC \tag{7}$$

In this study, $(b_{abs,\,BrC}/eBC)_{pri}$ was determined by the newly developed MRS method (Wu and Yu, 2016) using mass concentration of BC as a tracer (Wang et al., 2019a). In MRS calculation, the correlation ($R^2$) between measured eBC and estimated $b_{abs,\,Secondary\,BrC}$ was examined as a function of a series of hypothetical $(b_{abs,\,BrC}/eBC)_{pri}$. The $(b_{abs,\,BrC}/eBC)_{pri}$ with

the minimum correlation coefficients ($R^2$) between BC and $b_{abs,\,Secondary\,BrC}$ was assumed as the most statistically probable $(b_{abs,\,BrC}/eBC)_{pri}$ considering the independent variations between BC and $b_{abs,\,Secondary\,BrC}$.





### 2.4 Estimation of radiative forcing of BC and BrC

We estimated the direct radiative forcing ($\Delta F_R$) caused by BC and BrC at the top-of-atmosphere (TOA) based on forcing equations suggested by pervious study (Chylek and Wong, 1995), the modified wavelength-dependent version of the equation is given as below (Chen and Bond, 2010):

$$\Delta F_R = \int -\frac{1}{4}\frac{dS(\lambda)}{d\lambda}\tau_{atm}^2(\lambda)(1-F_c)[(1-a_s)^2 2\beta\tau_{scat}(\lambda)-4a_s\tau_{abs}(\lambda)]d\lambda \qquad (8)$$

where $S$ is the solar irradiance (W m⁻²), $\tau_{atm}$ is the atmospheric transmission (unitless), $F_c$ is the fractional cloud amount (unitless), $a_s$ is the surface reflectance (unitless), and $\beta$ is the backscatter fraction (unitless), and $\tau_{scat}$ and $\tau_{abs}$ are the aerosol scattering and absorption optical depths (unitless), respectively. Wavelength-dependent $S(\lambda)$ and $\tau_{atm}(\lambda)$ are derived from the ASTM G173-03 reference spectra (Chen and Bond, 2010). $F_c$ and $a_s$ are 0.6 and 0.19, respectively based on previous studies (Bond and Bergstrom, 2006; Wang et al., 2019b). $\beta$ is 0.29 (Charlson et al., 1992). Based on method in previous study (Wang et al., 2019b), $\tau_{scat}$ and $\tau_{abs}$ can be estimated as $\tau_{scat}$ ($\lambda$) = $b_{sca}$ ($\lambda$) × $H_{eff}$ and $\tau_{abs}$ ($\lambda$) = $b_{abs}$ ($\lambda$) × $H_{eff}$, respectively, where $b_{sca}$ ($\lambda$) and $b_{abs}$ ($\lambda$) are scattering and absorption coefficients, respectively, and $H_{eff}$ is effective height. The effective heights can be derived from the relationship between aerosol optical depth $\tau$ (=$\tau_{scat}$ + $\tau_{abs}$, available from the Aerosol Robotic Network data archive) and light extinction coefficient by CAPS. The detail results of $H_{eff}$ in four seasons are shown in Table S2.

The uncertainties of BC and BrC absorption $\Delta F_R$ for (including primary and secondary ones) were quantitatively determined through the use of Monte Carlo simulations. Note that the uncertainty was represented as one standard deviation (±1σ) or the coefficient of variation (CV, σ divided by the mean) expressed as a percentage. According to uncertainty propagation, the CV for $b_{abs, BC}$ ($\lambda$) as follows, $CV_{babs,BC(\lambda)} \approx \{ [(CV_{babs,BC, 880})^2+[CV_\alpha*\alpha*\ln(880/\lambda)]^2\}^{1/2}$, where $CV_{babs,BC, 880}$ and $CV_\alpha$ represent the uncertainty of measured absorption coefficient at 880nm (~ 25 %) and absorption Ångström exponent of pure BC (~ 10 %) (Lu et al., 2015; Bond et al., 2013; Lack and Langridge, 2013; Gyawali et al., 2009), respectively. The CV for $b_{abs, BrC}$ ($\lambda$) could be quantified as follows, $CV_{babs, BrC(\lambda)} \approx \{ [(CV_{babs,BrC, 370})^2+[CV_\beta*\beta*\ln(370/\lambda)]^2\}^{1/2}$ where $CV_{babs,BrC, 370}$ and $CV_\beta$ represent the uncertainties of BrC absorption coefficient at 370 nm ($CV_{babs,BrC,370} \approx \{ [(CV_{babs,total,370})^2+[ CV_\alpha*\alpha*\ln(880/370)]^2\}^{1/2} \approx 26$ %) and absorption Ångström exponent of BrC (fitting uncertainty ~ 10 %), respectively. Similarly, $CV_{babs, PriBrC, (\lambda)}$ and $CV_{babs, SecBrC(\lambda)}$ also could be quantified. The CVs were as the parameters for the Monte Carlo analysis, and 100000 simulations were conducted to evaluate uncertainties.

## 3 Results and discussion

### 3.1 Temporal variations of BC

Fig. 1a shows the annual variations of eBC, △eBC/△CO and eBC/PM₂.₅ during 2012-2020. The annual mean (±1σ) concentration of eBC was 5.54 ± 5.25 µg m⁻³ in 2012 (from August in 2012 to June in 2013) and decreased by 67 % (1.80 ± 1.54) in 2020. The Mann-Kendall trend test supported that the decrease in eBC from 2012 to 2020 was significant (Table S1). The annual mean concentration in 2020 was similar to that in Milan (Mousavi et al., 2019), lower than that in Xiamen (Deng et al., 2020), Shanghai (Wei et al., 2020), and Hefei (Zhang et al., 2015), yet higher than that in Nanjing (Jing et al.,



2019) and New York City (Rattigan et al., 2013). A similar reduction in CO by 68 % from 2012 to 2020 (Fig. S2) also indicated that the primary emissions from incomplete combustion reduced significantly in the past decade. Considering that different primary sources showed different emissions of BC and CO (Spackman et al., 2008; Derwent et al., 2001), we calculated $\triangle eBC/\triangle CO$ as the ratio of ($eBC-eBC_0$) and ($CO-CO_0$) which was widely used to identify the variations of BC sources (Kondo et al., 2006; Subramanian et al., 2010). $CO_0$ was determined as values of the 1.25 percentile of each year

and $eBC_0$ concentration was assumed as zero (Pan et al., 2011; Han et al., 2009). As shown in Fig. 1a, the annual mean values of $\triangle eBC/\triangle CO$ and $eBC/PM_{2.5}$ presented similarly decreasing trends, indicating a significant change in the structure of primary emission sources. Fig. 2 presents the monthly variations of eBC, $\triangle eBC/\triangle CO$ and $eBC/PM_{2.5}$. The eBC showed consistently seasonal patterns across different years with wintertime eBC almost twice that in summer mainly due to largely enhanced coal combustion emissions in heating season (Sun et al., 2018), consistent with the higher

$eBC/PM_{2.5}$ in wintertime. $\triangle eBC/\triangle CO$ presented pronounced seasonal variations. The highest values up to 12.0 ng m$^{-3}$ ppbv$^{-1}$ occurred in spring and fall likely due to the influences of biomass burning emissions (Pan et al., 2011; Han et al., 2009; Streets et al., 2003; Westerdahl et al., 2009; Spackman et al., 2008). However, the monthly average $\triangle eBC/\triangle CO$ became relatively constant after 2018, suggesting that the primary emission sources of eBC and CO were relatively stable after the five-year clean air action (2013 – 2017) in Beijing (Spackman et al., 2008).

As illustrated in Fig. 3, the mass concentrations of eBC were uniformly decreased in four seasons from 2012 to 2020 except for a small increase in 2018. According to the frequency distribution of $\triangle eBC/\triangle CO$ (Fig. S3), we found higher values of $\triangle eBC/\triangle CO$ in 2018 than 2019 and 2020 suggesting a change of primary emission after 2018. The largest decrease in eBC from 2014 to 2020 was observed in fall (78 %) with $\triangle eBC/\triangle CO$ and $eBC/PM_{2.5}$ decreasing 53 % and 3 %, respectively. The Mann-Kendall trend test (Table S1) also supported the long-term decreases in eBC and $eBC/PM_{2.5}$. The

eBC in spring decreased by 53 % from 2013 to 2020 with the similarly significant decrement in $\triangle eBC/\triangle CO$. The average $\triangle eBC/\triangle CO$ decreased from over 10 to below 5 in spring and fall, which suggested that the decreases of BC were mainly due to the reduced biomass burning emissions in the past eight years (Pan et al., 2011; Han et al., 2009; Streets et al., 2003; Westerdahl et al., 2009; Spackman et al., 2008). Different from the fall when BC emissions reduced more than other scattering pollutants, the values of $eBC/PM_{2.5}$ were relatively stable in spring. In comparison, the eBC and $\triangle eBC/\triangle CO$ in

summer were lower than those in other seasons except 2012, and they did not change substantially in recent years. This is likely due to relatively stable sources in summer, i.e., vehicle exhausts. Although the wintertime eBC decreased by more than 60 % from 2012 to 2019, the different source contributions to BC were relatively constant in winter over eight years as indicated by flat $\triangle eBC/\triangle CO$ (3 ~ 6 ng m$^{-3}$ ppbv$^{-1}$). One explanation is that coal combustion and biomass burning emissions in winter were reduced similarly in Beijing due to the promotion of clean fuels. Overall, the results above the

decreases in eBC during the last eight years in Beijing were mainly due to the changes in spring, fall and winter, and the reasons for the changes were different between winter and the other two seasons. In addition, more attention should be paid to the BC reduction in winter in the future based on the analysis of the four seasons.

Before 2015, the diurnal variation of eBC (Fig. 4) showed morning peaks which were mainly due to the traffic rush hours in four seasons. After the "China 5" standard applying nationally and eliminating 5 million old vehicles in China (Zhang

et al., 2019), the morning peaks of eBC disappeared. Instead, they eBC presented similar and pronounced diurnal variations





during four seasons characterized by the lowest mass concentrations in the afternoon due to high mixing layer height (MH) and low emissions, consistent with previous studies in Beijing (Xie et al., 2019). Due to deeper developments of boundary layer in spring and summer, the lowest values occurred during 15:00 ~ 17:00 which were later than those in fall and winter. Comparatively, ubiquitously higher concentrations of eBC in early morning resulted from a synergetic effect of shallow

boundary layer height and high emissions from heavy-duty vehicles and diesel trucks that are allowed to enter the city only between 23:00 and 6:00. Consistently, the diurnal cycles of $\triangle eBC/\triangle CO$ presented the highest values at 2:00 ~ 6:00 before 2019 due to the differences in vehicle emissions throughout the day. For example, previous studies found that CO is emitted primarily from gasoline vehicles while BC is dominated from diesel trucks and heavy-duty vehicles (Kondo et al., 2006). We also observed the decreases in $\triangle eBC/\triangle CO$ and eBC from 2013 to 2019 were more significant at night, highlighting

that the reductions of diesel truck and heavy-duty vehicle emissions at night contributed significantly to the decreases of BC in Beijing. Differently, the diurnal cycles of $\triangle eBC/\triangle CO$ in winter were less pronounced than other seasons (Fig. 4h), indicating that the sources of BC were relatively stable during heating period although the mass concentrations decreased. Particularly, we found that the diurnal variations of both $\triangle eBC/\triangle CO$ and eBC in 2020 were less pronounced during four seasons. One reason was likely due to the fact that primary emissions e.g., vehicle emissions, were significantly reduced

due to the influences of COVID-19. According to the diurnal variations, we found even though diesel vehicle emission reduction contributed much to the whole BC decrease, its emission should also be controlled especially in summer and fall.

Fig. 5 shows the bivariate polar plots of eBC during four seasons. In general, higher concentrations of eBC occurred in the region with low WS ($< 2$ m s$^{-1}$), while lower concentrations often occurred in the region with high WS from the northwest during all seasons. In summer, the eBC presented similar distribution with high concentration in the middle and also the

region to the south and southeast except 2012, suggesting the important contributions from both local emissions and regional transport. Different from the summer, high concentrations of eBC occurred dominantly in a small region close to the sampling site during other three seasons, suggesting the dominant source contributions from local emissions. However, the regional transport from the south and southeast was also found to play an important role, e.g., spring 2015 and 2020, and winter 2017. It's interesting to note that the eBC from the southwest with low WS decreased significantly over eight

years in fall while it still exceeded 3 µg m$^{-3}$ from the southeast in the fall of 2019. These results indicate that the source regions of eBC can be substantially different in different years depending on meteorology. By comparing with the seasonal variation of $\triangle eBC/\triangle CO$, we inferred that biomass burning emissions from the southwest and regional transport from the southeast are two important non-local sources of eBC in Beijing. In conclusion, in the process of implementing air pollutant reduction actions, the synergistic control of the surrounding areas of Beijing should not be neglected as well.

## 3.2 Temporal variations of aerosol optical properties

Fig. 1b presents annual variations of $b_{ext}$, SSA and MEE. The annual mean ($\pm 1\sigma$) of $b_{ext}$ was decreased by 47 % from 2012 to 2020, while that of SSA was increased from $0.79 \pm 0.11$ to $0.88 \pm 0.06$. Such an increase in SSA could likely shift the radiative forcing from positive to negative (Hansen et al., 1997; Lee et al., 2007). Besides SSA, MEE is also a key factor reflecting the responses of atmospheric light properties to aerosol composition changes. In particular, the annual mean

MEE and SSA increased despite the decreases in eBC and $b_{ext}$ in the past decade indicating that scattering aerosol species





played more important roles than absorbing aerosol species in radiative forcing. This change is consistent with the findings of previous studies showing increased contributions of high scattering ammonium nitrate in fine particles (Y. Huang et al., 2013; Lei et al., 2020). The seasonal variations of SSA and MEE showed generally higher values in winter and lower values in summer (Fig. S4). Such seasonal trends are overall similar to those of eBC/PM$_{2.5}$ and eBC (Fig. 2), indicating
that non-BC aerosol species in winter appeared to have higher scattering efficient than those in summer.

Fig. 6 shows that the seasonal average SSA presented similar increasing trends during all seasons indicating a more effective controls of absorbing aerosol (i.e., eBC) than scattering components during the last decade. This is consistent with recent studies showing larger reductions in primary aerosol species than secondary species as responses to emission controls (Sun et al., 2020b). The most significant increase of SSA was observed in fall from $0.75 \pm 0.12$ to $0.87 \pm 0.07$
during 2012-2020, followed by summer from $0.77 \pm 0.12$ in 2012 to $0.88 \pm 0.06$ in 2020. The highest seasonal average SSA ($0.88 \pm 0.06$) was observed in summer 2020, which is close to that during the COVID-19 outbreak in spring 2020, yet was much higher than $0.82 \pm 0.05$ observed in North China Plain in 2009 (Ma et al., 2011). We also noticed that the increase in SSA was becoming smaller over past eight years indicating that the relative contributions of light absorbing and scattering components became relatively stable as the progress of clean air action. Fig. 7 also shows the seasonal average
of $b_{ext}$ and MEE over past nine years. The MEE increased mostly by more than 43 % in summer from 2012 to 2020 although $b_{ext}$ decreased ubiquitously during all seasons, most notably in the fall from 432 Mm$^{-1}$ in 2014 to less than 140 Mm$^{-1}$ in 2020 ($\sim$ 68 %). Comparatively, $b_{ext}$ was relatively stable at 230 Mm$^{-1}$ in spring before 2019 and decreased substantially by 40 % in 2019 due to significant reductions in fine particles. Although $b_{ext}$ was comparable in springs of 2019 and 2020, SSA was increased by 8 %. These results suggest the aerosol composition has also played an important role in changing
aerosol optical properties. For example, higher SSA in spring 2020 resulted from larger reductions in primary emissions e.g., absorbing eBC, than scattering secondary aerosol due to the decreases in anthropogenic emissions during the COVID-19 lockdown. The increase of MEE from 2.6 to 3.6 m$^2$ g$^{-1}$ also suggested a significant change in scattering aerosol composition from 2019 to 2020, consistent with the results in a previous study (Lei et al., 2020). Compared with spring, $b_{ext}$ decreased by 60 % in summer from 2012 to 2015 and then gradually increased afterwards. Similarly, $b_{ext}$ also showed
a sharp decrease of 60 % in winter from 2014 to 2017 and after that it continuously increased to > 200 Mm$^{-1}$ in 2019. Considering the increased SSA yet relatively constant mass concentrations of eBC, we inferred that the increased light extinction in winter was mainly caused by scattering aerosols that can vary substantially in different years due to the changes in meteorological conditions (Zhou et al., 2019). Overall, the results in this study clearly demonstrate the responses of aerosol optical properties to the changes in aerosol composition since clean air action in 2013. In summary, seasonal
variations of SSA and MEE indicates that scattering aerosols will become a new challenge affecting atmospheric visibility in all seasons.

The diurnal cycles of $b_{ext}$ and SSA in four seasons are shown in Fig. 7. The diurnal variations of SSA were similar and pronounced during all seasons that were characterized by the peaks at afternoon, consistent with previous studies in Beijing (Xie et al., 2019; Han et al., 2017). Before 2015, SSA presented an obvious valley during 7:00 ~ 9:00 mainly due to the
increased BC concentrations and contributions, and the valley was much smaller after 2015 due to the improvement of vehicle emission standards and the reductions in vehicle emissions. After the early morning, SSA increased presented the





highest values during 12:00 ~ 13:00. The major reason is the reduced eBC emissions during daytime and enhanced emission and photochemical production of secondary scattering aerosols (Han et al., 2017).

$b_{ext}$ also presented similar diurnal variations over nine years which were characterized by higher values at night and lower values during daytime. One of the major factors driving the diurnal variations is the evolution of boundary layer height (Han et al., 2017; Xie et al., 2019; Han et al., 2015). As a response, $b_{ext}$ reached the minimum at 12:00 ~ 14:00 in fall and winter whereas it occurred during 16:00 ~ 18:00 in spring and summer due to a deeper mixing convection in late afternoon. In fall, the value of $b_{ext}$ decreased while SSA increased at nighttime from 2012 to 2019, indicating that the reduction of BC at night had a significant impact on the decrease of $b_{ext}$. Note that $b_{ext}$ increased by more than 62 % in winter from 2018 to 2020 as discussed above, yet the reasons causing the increased $b_{ext}$ were different according to the diurnal variations. For example, the diurnal variations of $b_{ext}$ in winters of 2017 and 2018 suggested that increased $b_{ext}$ was mainly due to simultaneously enhanced absorbing and scattering aerosols at night, consistent with relatively similar diurnal patterns of SSA in the two winters. However, the relative consistency of $b_{ext}$ values from 20:00 to 4:00 in 2018 and 2019 indicated that the increase in seasonal average of $b_{ext}$ from 2018 to 2019 was mainly due to the increase in scattering aerosols during the daytime when the eBC mass concentration was relatively stable. Overall, the reduced eBC and increased scattering aerosols together resulted in the increase in seasonal average of SSA by more than 4 % from 2018 to 2019.

### 3.3 Temporal variations of light absorption of BrC

Fig. 8a shows the seasonal variations of BrC absorption and AAE during 2017-2020. The seasonal average absorption of BrC was the highest during winter which was approximately twice that in spring and fall, and five times higher than that in summer. Consistent with the seasonal variations of eBC, the absorption of BrC in 2018 was generally higher than other years mainly due to the increased biomass burning emissions. The lowest AAE ubiquitously occurred in summer while the highest value up to 1.5 occurred in winter, consistent with previous studies (Xie et al., 2020; Xie et al., 2019). Despite the stronger absorption in fall than spring in 2018 and 2019, the AAE was similar indicating the similar emission sources of BrC in the two seasons (Ran et al., 2016). Note that AAE was up to 1.39 in 2020 and showed a higher frequency at AAE > 1.3 in spring (Fig. 9), suggesting that the emissions with high combustion efficiency (e.g., traffic) decreased much more than the low efficiency sources (e.g., biomass burning) during the COVID-19 lockdown in Beijing. As illustrated in Fig. 9, the distribution of AAE in summer mainly concentrated in the range of 1.0 - 1.3 due to the low source emissions BrC than other seasons, particularly primary coal combustion and biomass burning emissions. However, we found a change in AAE distribution in summer 2020, which was characterized by a higher frequency at AAE > 1.3 suggesting a stronger BrC absorption. Further analysis showed that such a change was mainly due to the enhanced contribution from secondary BrC. Compared with summer, the AAE distribution was relatively stable in fall and winter, and the distribution range of ~ 1.2 - 1.9 in winter was overall higher than that in other seasons

As shown in Fig. S5, the diurnal variations of BrC absorption were similar to eBC with generally higher values at nighttime except in summer during 2018 – 2020. This result indicated that the primary emissions related BC were also the main sources of BrC in spring, fall and winter. In comparison, the diurnal variations of BrC absorption were largely different from eBC in summer, possibly due to the fact that BrC was significantly influenced by secondary organic aerosols. The





diurnal variation of BrC absorption in summer 2020 was different from previous years while the variations of eBC did not change significantly, supporting the increased contribution of secondary aerosol to BrC. Generally, the AAE showed a minimum at night followed by a daytime increase from 8:00 to 12:00 during four seasons (Fig. S5), suggesting that
photochemical production contributed dominantly to the BrC formation during daytime.

By using the minimum $R$ squared method (MRS) (Wu and Yu, 2016), we estimated the primary and secondary BrC absorptions in each month during 2017-2020. As shown in Fig. S6, the monthly variations of BrC absorption were similar and pronounced which were characterized by high values in January and low values in July. Despite this, the primary BrC absorption decreased gradually from 2017 to 2020, mainly due to the decreased emissions of biomass burning and coal
combustion. We further explored the seasonal variations of primary and secondary BrC. As shown in Fig. 8b, BC dominated ultraviolet light absorption at 370 nm during four seasons with the highest contribution being in summer (~ 85 %) and the lowest in winter (~ 60 %). One reason is because BrC from biomass burning and coal combustion in summer was small, consistent with lower AAE in summer than other seasons (Fig. 9). Note that the average contribution of BrC to the total absorption in summer increased to 16 % from 2018 to 2020 due to enhanced photochemical production associated with
stronger atmospheric oxidation capacity. Comparatively, the contributions of primary BrC to the total absorption decreased from 75 % to 50 %. The contributions of BrC absorption were comparable in spring and fall, accounting for 25-30 %. Due to the decreased primary emissions and enhanced secondary production during the COVID-19, we found that the contribution of BrC absorption was increased by more than 7 % in spring from 2018 to 2020. In comparison, the contributions of BrC were larger than 40 % in winter with slightly downward trends in past three years. The declines of
primary emissions might be an explanation, mainly due to the replacement of coal to natural gas for residential heating in recent years.

Although the contributions of BrC absorption to the total absorption were relatively stable during four seasons from 2018 to 2020, the relative contributions of primary and secondary BrC changed significantly (Fig. 8c). Overall, the primary BrC was much higher than secondary BrC, yet showing decreasing trends from 2017 to 2020 except in spring. While the primary
BrC contributed more than 75 % to the total BrC in fall and winter, they reached the minimum in summer (50-75 %). The contribution of summertime primary BrC decreased by more than 25 % from 2018 to 2020, and that of secondary BrC increased up to 50 % to the total BrC in summer 2020 with an increase in AAE to 1.2. Given that eBC was continuously decreased in summer in past three years, the increases in AAE were mainly due to larger secondary BrC production from photochemical reaction. We also observed a large increase in secondary BrC in winter from 2018 to 2020. While the
secondary BrC was negligible in 2018, the contribution increased to ~ 25 % in 2020, suggesting that secondary production of BrC became more important in winter, which is consistent with the continuous increase SOA in winter in recent years (Lei et al., 2020). Similar increases in secondary BrC were also observed in fall.

### 3.4 Direct radiative forcing of BC and BrC

As shown in Fig. 10, the annual mean $\Delta F_R$ caused by BC was about +3.00 W m$^{-2}$ in 2012 (from August in 2012 to June in
2013), close to that previously reported in north China (Yang et al., 2017). However, $\Delta F_R$ decreased substantially by 64 % (+1.09) in 2020, suggesting that the BC radiative forcing was largely reduced during the last decade. Previous studies (Ding



et al., 2016) had shown that the aerosol-boundary layer feedback to unit quantity of BC will be lower in higher aerosol loading case as solar radiation weakened. In our study, such decrease of BC radiative effect likely contributed much to quick improve the air quality. And the relatively lower $\Delta F_R$ caused by BC in recent year could facilitate the dispersion of

pollutants in the boundary layer, which in turn will maintain air pollution at a low level. The seasonal variation of BC $\Delta F_R$ (Fig. S7) suggested the largest decrease in summer and fall. In addition, we noticed that the BC $\Delta F_R$ was relatively stable in each season from 2019 and 2020, consistent with the small changes in eBC concentrations. We also estimated the radiative effects of BrC. As shown in Fig. 10, BrC $\Delta F_R$ decreased by 43 % from +0.30 in 2018 to +0.17 W m$^{-2}$ in 2020. However, such a value was much higher than the global mean (+0.04 ~ 0.11 W m$^{-2}$ ) (Feng et al., 2013). The scattering

radiative forcing of BrC was estimated at -1.00 ~ -1.65 W m$^{-2}$. The absorbing radiative forcing of BrC led to ~18 % reduction in the amount of negative radiative forcing caused by BrC scattering compared to results from the non-absorbing assumption. The seasonal variation of BrC $\Delta F_R$ (in Fig. S7) showed a large decrease during all seasons from 2018 to 2019. However, compared with 2019, the BrC $\Delta F_R$ became stable in summer 2020 which was different from the decreases in spring and fall. We also estimated the primary and secondary BrC $\Delta F_R$. Primary BrC $\Delta F_R$ was approximately +0.16 W m$^{-2}$

in 2020 decreasing by 41 % compared with 2018 (+0.27 W m$^{-2}$). Such a value was higher than the global average of radiative forcing (+0.11 W m$^{-2}$) from POA (Lu et al., 2015). Compared with primary BrC, the secondary BrC $\Delta F_R$ was generally small yet showing an increase from +0.005 W m$^{-2}$ in 2019 to +0.016 W m$^{-2}$ in 2020. The probability distributions of $\Delta F_R$ for BC and different types of BrC are shown in Fig. S8. The uncertainties of BC and BrC absorption $\Delta F_R$ are comparably about 27 ~ 28 %. And the uncertainties for primary and secondary BrC absorption $\Delta F_R$ are about 32 % and 43

%, respectively.

## 4 Conclusions

Nine-year measurements of eBC and light extinction coefficient in Beijing were analyzed in this study. Our results showed that the annual mean eBC concentration decreased by 67 % from 5.54 µg m$^{-3}$ in 2012 to 1.80 µg m$^{-3}$ in 2020, and the decreases dominantly occurred at nighttime suggesting an effective control of primary emissions due to clean air action

since 2013. $b_{ext}$ showed similar reductions by 47 % from 2012 to 2020. We also observed a pronounced seasonal variation in $\triangle$eBC/$\triangle$CO with high values in spring and fall, and a gradual decrease in recent years, indicating a significant change in primary sources. As a response of the changes in primary and secondary aerosols, SSA increased substantially from 0.79 ± 0.11 in 2012 to 0.88 ± 0.06 in 2020, and it presented similar increasing trends during all seasons. These results highlight increasingly important role of scattering aerosol in radiative forcing. Similarly, the seasonal average MEE increased

gradually from 2012 to 2020, and the increase was most significant in summer by more than 43 %. The increased MEE explained the fact that PM$_{2.5}$ decreased substantially after clean air action, while the visibility did not show similar improvements as PM$_{2.5}$.

We further analyzed the changes in BrC during 2018 – 2020. The BrC absorption presented the pronounced seasonal variation with the highest value in winter, after quantifying the primary and secondary BrC, we found that the primary

emissions co-emitted with BC were the main sources of BrC during most seasons while the secondary BrC was also important in summer. In particular, the contribution of secondary BrC to the total BrC showed a large increase in summer,



and it was up to 50 % in summer 2020. These results indicated the BrC from secondary formation played an increasing role in the absorption at 370 nm during 2018 – 2020 in Beijing. By estimating the direct radiative forcing caused by absorbing aerosols, we found that the annual mean BC $\Delta F_R$ decreased by 64 % from +3.00 W m$^{-2}$ in 2012 to +1.09 W m$^{-2}$ in 2020, and that of BrC decreased from +0.30 to +0.17 W m$^{-2}$ during 2018 - 2020. Considering that the BC-induced aerosol and boundary layer feedback plays an important role in severe haze formation, the decreases in BC and radiative forcing would weak the interaction between aerosol and boundary layer, and help mitigate air pollution.

*Data availability.* The data in this study are available from the authors upon request (sunyele@mail.iap.ac.cn).

*Author contributions.* YS and JS designed the research. JS, WZ, CX, CC, and TH conducted the measurements. JS, ZW, WZ and CC analyzed the data. CW, QW, ZL, JL, PF and ZiW reviewed and commented on the paper. JS and YS wrote the paper.

*Competing interests.* The authors declare that they have no conflict of interest.

*Acknowledgements.* This work was supported by the National Natural Science Foundation of China (42061134008, 9204430003).



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

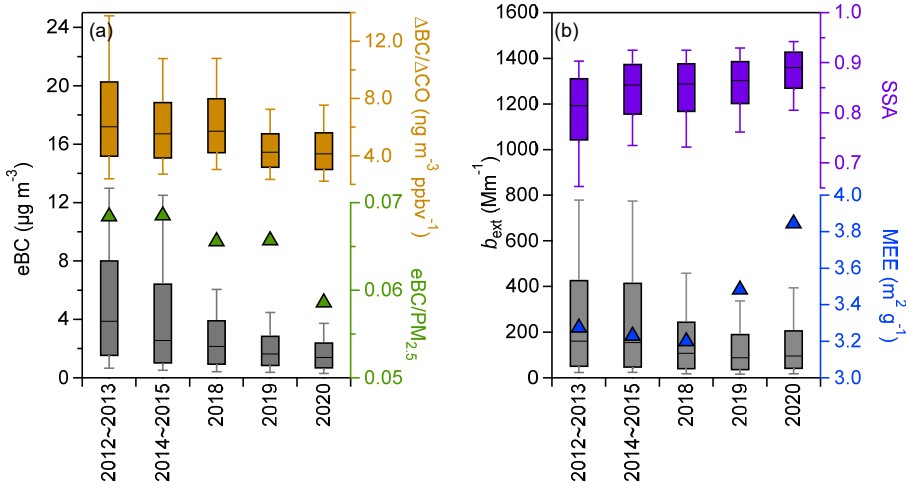

Fig. 1. Annual variations of (a) eBC, △eBC/△CO, eBC/PM2.5, (b) $b_{ext}$, SSA and MEE. The median (horizontal line), mean (markers),25th and 75th percentiles (lower and upper box), and 10th and 90th percentiles (lower and upper whiskers) are also
shown, same as below.



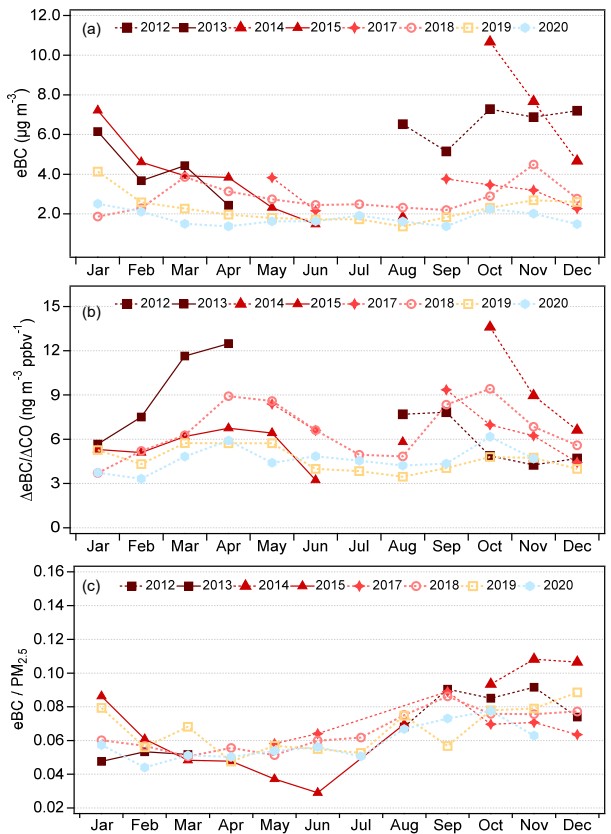

**Fig. 2. Monthly variations in (a) eBC, (b) △eBC/△CO and (c) eBC/PM2.5.**

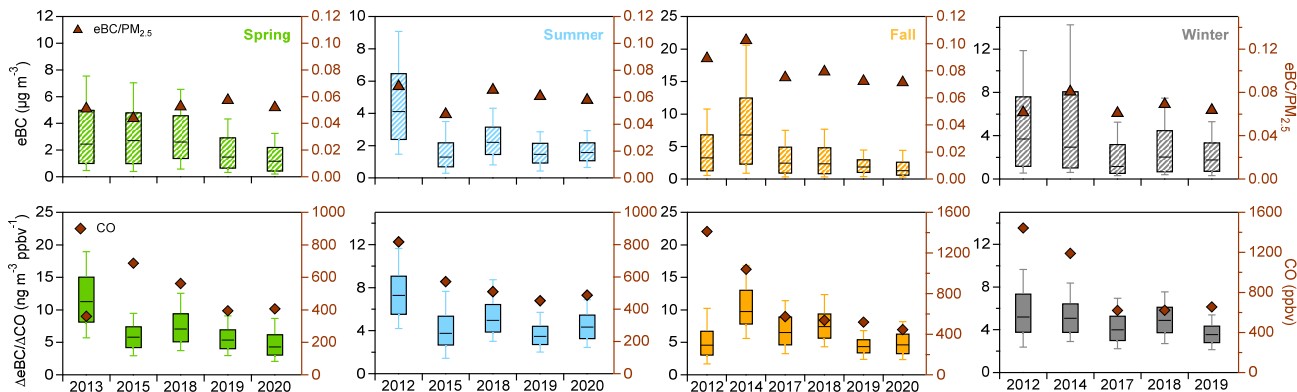

**Fig. 3. Seasonal variations in eBC, eBC/PM2.5, △eBC/△CO and CO.**





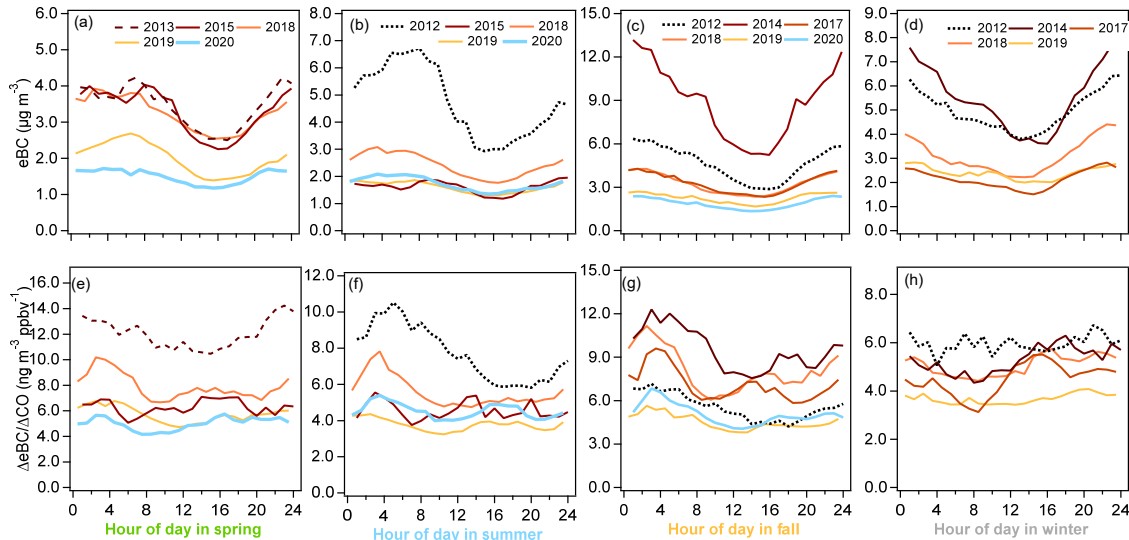


**Fig. 4. Diurnal variations of eBC and △eBC/△CO for spring, summer, fall and winter time in different years.**







**Fig. 5. Bivariate polar plots for hourly eBC mass concentration in the four seasons over nine years.**




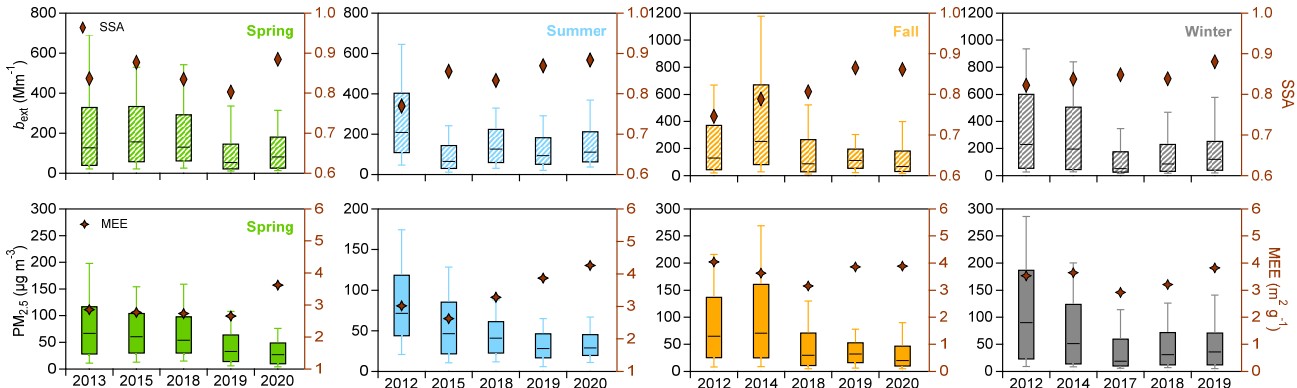

**Fig. 6. Seasonal variations in $b_{ext}$, SSA, PM$_{2.5}$ and MEE.**

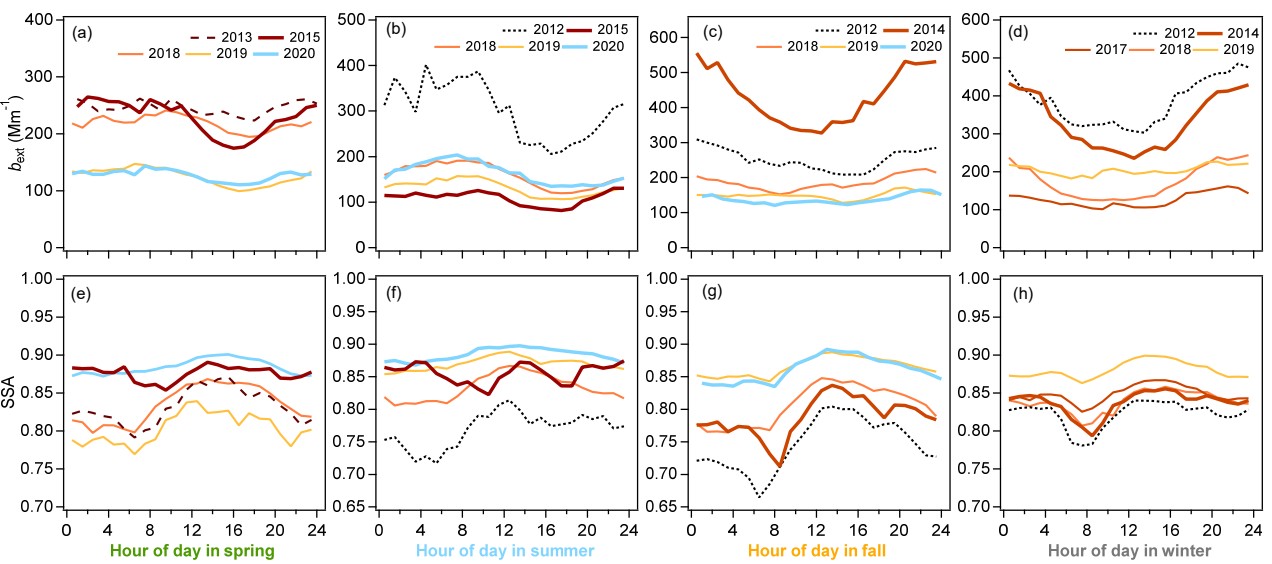

**Fig. 7. Diurnal variations of $b_{ext}$ and SSA for spring, summer, fall and winter time in different years.**




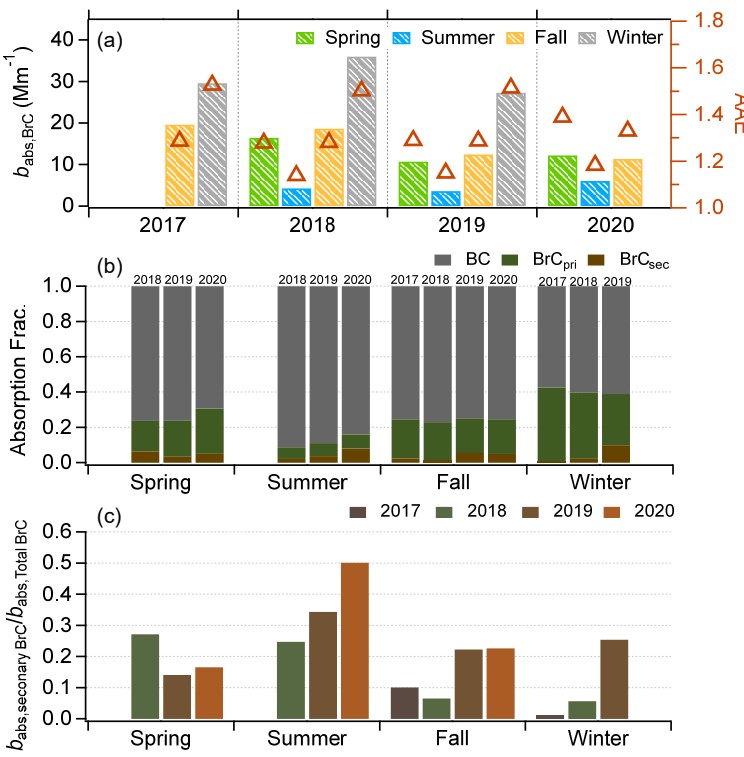

**Fig. 8.** Seasonal variations of (a) $b_{abs, BrC}$, AAE, percentage contribution of (b) absorbing components to the absorption coefficient and (c) the proportion of $b_{abs, Secondary BrC}$ in BrC absorption coefficient at 370nm from 2018 to 2020.





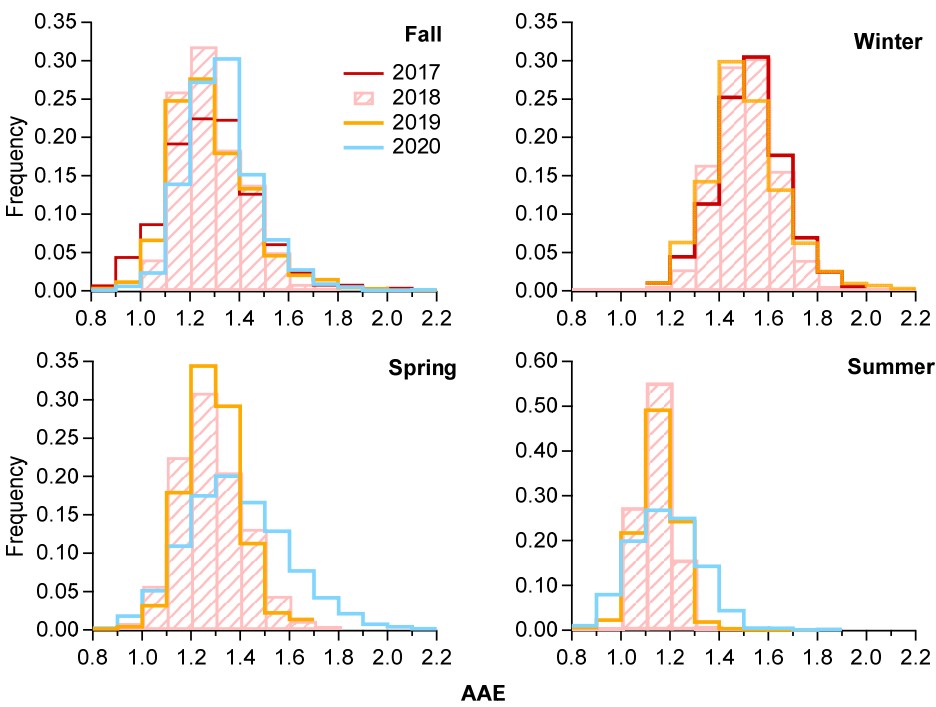


**Fig. 9. The frequency distributions of AAE in four seasons.**

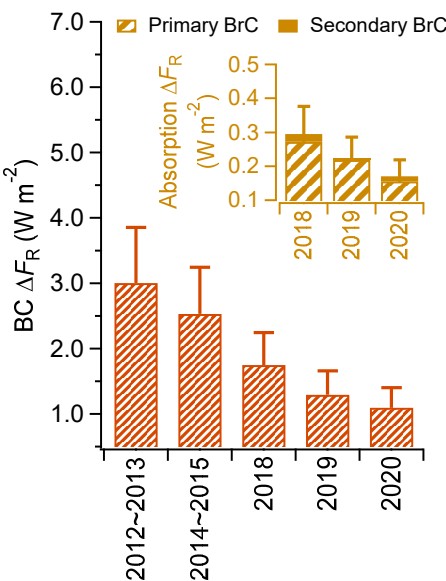

**Fig. 10. Temporal variations of the annual mean $\Delta F_R$ caused by BC, primary BrC and Secondary BrC.**