# Peer review of "Measurement report: Long-term changes in black carbon and aerosol optical properties from 2012 to 2020 in Beijing, China"

_Atmospheric Chemistry and Physics, 2021_

## Author Response (AR1)

**Response to Reviewers' comments**

We are thankful to the two reviewers for their thoughtful and constructive comments that help improve the manuscript substantially. We have revised the manuscript accordingly. Listed below is our point-to-point response in blue to each comment that was offered by the reviewers.

**Response to Reviewer #1**

General comments

This paper presents data from about a 9 year period showing a general decrease in black carbon aerosol in Beijing and the implications of that decrease for single scattering albedo and radiative forcing. Overall, this work provides a useful study for publication, but a number of items should be addressed in the final draft.

First, carefully proofread the paper for grammar, proper word choice, etc. I list a few things below, but I do not list all the errors in the current draft.

Great thanks to the reviewer to point out the grammar and word choice errors. We have checked the manuscript thoroughly and corrected the errors.

Second, the MRS method is newly developed and not cited much; thus, I think it would be appropriate to provide a bit more detail in your paper on how this method separates primary and secondary BrC. If necessary, provide some graphs in the Supplemental.

We assumed that the BC particles were from primary emissions, while $b_{abs,\ Secondary\ BrC}$ was from secondary organic aerosols that were mainly formed through oxidation of volatile organic compounds. To meet the assumption that BC and $b_{abs,\ Secondary\ BrC}$ were from different sources and independent, the minimum $R^2$ for $b_{abs,\ Secondary\ BrC}$ vs. BC was the target of this analysis.

The hourly $b_{abs,\ BrC}$ and eBC measurement data in each month are used to calculate the $(b_{abs,\ BrC}/eBC)_{pri}$ by using the minimum $R^2$ method. Fig. R1 shows an example of the results in fall in 2020. We then estimate the average $b_{abs,\ Primary\ BrC}$ in each month by eq. (1) where [eBC] refers to the average mass concentration of eBC in this month. The average $b_{abs,\ Secondary\ BrC}$ was determined as the difference between the total $b_{abs,\ BrC}$ and $b_{abs,\ Primary\ BrC}$. In the revised manuscript, we presented the result of MRS for each month in Figure S8.

$$b_{abs,\ Primary\ BrC} = (b_{abs,\ BrC}/eBC)_{pri} * eBC \qquad (1)$$

[Figure]

Fig. R1. $(b_{abs, BrC}/eBC)_{pri}$ determination by MRS at 370nm in September, October, November in 2020. The red line represents the correlation coefficient ($R^2$) between hypothetical $b_{abs, Secondary BrC}$ and eBC mass as a function of $(b_{abs, BrC}/eBC)_{pri\_h}$. The shaded area in light tan represents the frequency distribution of observed $(b_{abs, BrC}/eBC)_{pri}$. The dashed green line is the cumulative distribution of observed $(b_{abs, BrC}/eBC)_{pri}$.

Following the reviewer's suggestions, we added Fig. R1 as Fig. S2 in the revised manuscript and expanded the discussion on the method in Section 2.3:

"In this study, $(b_{abs, BrC}/eBC)_{pri}$ was determined by the newly developed MRS method (Wu and Yu, 2016) using mass concentration of BC as a tracer (Wang et al., 2019a). In MRS calculation, the correlation coefficients ($R^2$) between measured eBC and estimated $b_{abs, Secondary BrC}$ was examined as a function of a series of hypothetical $(b_{abs, BrC}/eBC)_{pri}$. The $(b_{abs, BrC}/eBC)_{pri}$ with the minimum correlation coefficients ($R^2$) between BC and $b_{abs, Secondary BrC}$ was assumed as the most statistically probable $(b_{abs, BrC}/eBC)_{pri}$ considering the independent variations between BC and $b_{abs, Secondary BrC}$. Based on this method, we first determined the monthly $(b_{abs, BrC}/eBC)_{pri}$ with an example of analysis of three months in the fall of 2020 (Fig. S2). The $b_{abs, Secondary BrC}$ was then determined as the difference between the total $b_{abs, BrC}$ and $b_{abs, Primary BrC}$."

Likewise, I do not understand the Monte Carlo simulations of uncertainties on radiative forcing. Please explain this process better.

The uncertainty of radiative forcing is estimated through Monte Carlo simulations. This method has been widely used in previous studies (Lu et al., 2015; Wang et al., 2019b). The major source of the uncertainty is from the measurements. The uncertainties of the measurements can be represented as one standard deviation (σ) or the coefficient of variation (CV = σ divided by the mean) expressed as a percentage. We first estimated the CV(λ) of measured $b_{abs, BC}(\lambda)$ as a function of the λ and $b_{abs, BC (880)}$. Then we got $b_{abs, BC}(\lambda)$ by using a random function following the normal distribution with μ= $b_{abs, BC}(\lambda)$, and σ=CV(λ) *μ for 100 000 times. Running the radiative forcing model repeatedly by using those $b_{abs, BC}(\lambda)$, we got 100 000 RF values, and the standard deviation could be considered as the uncertainty of radiative forcing.

We revised the text as:

Then we applied normal distributions for measured data with uncertainties provided by the calculated CVs and 100 000 simulations by Monte Carlo analysis. After running the radiative forcing model repeatedly, the standard deviation of 100 000 RF values could be considered as the uncertainty of radiative forcing.

Regarding data interpretation, please comment on the fact that in Figure 1, the major downshift in eBC concentration corresponds to the switch from one aethalometer to the other. Did the old model have a higher uncertainty or more noisy signal? Please detail the extra correction you refer to in line 93, as this correction plays a role in how these data look on Fig 1.

Drinovec et al. (2015) found that the loading of the filter spot results in a linear reduction of the instrumental sensitivity for the old version of Aethalometers e.g., AE22. As a result, they developed a dual-spot approach and incorporated into the new Aethalometer model AE33 which was a new real-time loading effect compensation algorithm based on the dual-spot measurements. Compared with AE22, AE33 shows high performance and provides reliable eBC measurements. In our study, we conducted a three-day inter-comparison between AE33 and AE22 (Han et al., 2017). The results showed that AE22 exhibited a high correlation coefficient with AE33, supported by $R^2$ = 0.99. However, the slope of AE33 vs. AE22 was about 1.38, indicating that AE22 underestimated the eBC mass concentration about 38%. Thus, we corrected the eBC measurements by AE22 from August 2012 to December 2014 according to the coefficient (1.38). Also, we have described the correction in detail in the revised manuscript. It now reads:

"Because the new version of AE33 using "dual-spot" technique can provide more reliable measurements by better correcting the filter-based loading effects (Drinovec et al., 2015), we further corrected the eBC measurements of AE22 according to a parallel measurement between AE33 and AE22 ($R^2$ = 0.99, slope = 1.38) (Han et al., 2017)."

It is quite interesting that the eBC and CO both drop by nearly exactly the same percentage from 2012 to 2020, and yet the emission ratio of delta eBC / delta CO also changes, thus indicating a change in primary emission source. My question is in regards to how you calculate the emission ratio - why do you use a mean or initial eBC value of zero while you don't for CO? Provide justification in the text. Also, if there is any uncertainty to performing the calculation this way, what effect would that have on your final emission ratio trends?

We thank the reviewer's comments. The described decreases in CO and eBC are not from the same period, e.g., CO is from January 2012 to 2020 and the eBC is from August 2012 to 2020. We thank the reviewer for pointing this out. To make it clearer, we have changed the graph of CO in the supplement to make the time period the same

as eBC (Fig. R2). According to the figure, the decrease in CO was about 56% from 2012 (August 2012 to May 2013) to 2020, which was lower than that of eBC (~71%) at the same time.

Previous studies suggested that almost no natural sources of BC in clean background except biomass burning or wild fires, and $eBC_0$ concentration was assumed as zero while the 1.25 percentile value of $CO_0$ regarded as its clean background concentration (Pan et al., 2011; Subramanian et al., 2010). The reason we did not use zero for $CO_0$ is that CO has a life time of 30–90 days in the troposphere, therefore it can be transported a long distance causing a regional background value. Comparatively, BC has a much shorter lifetime, approximately a week because it can be removed from the atmosphere through wet (i.e., in precipitation) and dry deposition to the Earth's surface. Another mainstream approach indicated that both $CO_0$ and $BC_0$ were determined as the averages of the 1.25 percentiles during measurements (Kondo et al., 2006; Han et al., 2009). As shown in Figure R3, the difference of the annual average $\triangle eBC/\triangle CO$ between the two methods is below 1 ng m$^{-3}$ ppbv$^{-1}$ with a relative uncertainty ~ 3 - 9%. Most importantly, the annual trends of $\triangle eBC/\triangle CO$ are extremely similar indicating that calculation of $\triangle eBC/\triangle CO$ with different methods would not affect our conclusion.

Following the reviewer's comment, we expanded the description in the revised manuscript:

The background concentration of CO ($CO_0$) was determined as the average of 1.25 percentile in each year, and that of eBC ($eBC_0$) was assumed as zero considering the negligible natural sources of BC in the clean background except biomass burning and wild fires (Pan et al., 2011; Han et al., 2009) and the short lifetime in the atmosphere (Bond et al., 2013).

[Figure]

Fig. R2. Annual variation of CO concentration. The median (horizontal line), mean (square), 25th and 75th percentiles (lower and upper box), and 10th and 90th percentiles (lower and upper whiskers) are also shown.

[Figure]

Fig. R3. Comparison of the trends of $\triangle eBC/\triangle CO$ that were calculated with different background concentrations.

Talk more about the possible emission sources and how they match to your emission ratio calculation.

BC and CO are both from the emissions of incomplete combustion. $\triangle eBC/\triangle CO$ can change significantly depending on fuels and burning conditions, therefore, it is challenging to accurately quantify the different emission source contributions to the changes in $\triangle eBC/\triangle CO$, yet it can be very useful for source diagnostics. In our study, we mentioned the $\triangle eBC/\triangle CO$ values from biomass burning, coal combustion, gasoline vehicles, diesel trucks and heavy-duty vehicles, which showed relatively consistent results from previous studies. Following the reviewer's suggestions, we cited more studies to support our conclusions and provided more sufficient BC source emissions , e.g., "$\triangle eBC/\triangle CO$ (4 ~ 6 ng m$^{-3}$ ppbv$^{-1}$, representing typical coal combustion emission in winter (Wang et al., 2015)", and "CO is emitted primarily from gasoline vehicles which showed the low $\triangle eBC/\triangle CO$ about 3 ng m$^{-3}$ ppbv$^{-1}$ while BC is dominated from diesel trucks and heavy-duty vehicles (Kondo et al., 2006; Han et al., 2009)".

You mention biomass burning as a significant source of BC, and we see in the data that BC emissions are decreasing over time.   Can you elaborate on what types of biomass burning: wildfires, prescribed fires, residential heating, etc. are prevalent in the study area or may be transported over the site?

The major type of biomass burning in June and October are agriculture burning, while it is residential burning during other months. Because biomass burning is banned in the city and rural area of Beijing, it is dominantly from regional transport to the south and southwest. The decrease in BC from biomass burning was mainly due to the policy in recent years that prohibits agriculture burning in North China Plain.

Why does the change in primary emission come in 2018, when the clean action plan was for 2013-2017?   Did nothing happen until the end of the 5 year plan?

Thank the reviewer's comment. After clean air action from 2013 to 2017, the government implemented "Three-Year Action Plan for Blue Sky Defense" from 2018 to 2021 (www.gov.cn/zhengce/content/2018-07/03/content_5303158.htm) with continuous emission controls. Such a policy resulted in a further decrease in eBC and then gradually remained stable at lower levels from 2019 to 2020. It should be noted that meteorological conditions also played an important role in affecting eBC concentrations. To highlight the significant changes due to the Clean Action Plan, we changed the description as follows:

"Especially, the mass concentration of eBC decreased by more than 63% and 44% from 2014 to 2017, and even up to75% in summer from 2012 to 2015, and the $\triangle$eBC/$\triangle$CO changed differently in different seasons. These results indicate a significant response of black carbon aerosol to the "Atmospheric Pollution Prevention and Control Action Plan". Although a temporary increase in eBC and $\triangle$eBC/$\triangle$CO (Fig.3 and Fig. S4) in 2018 suggested a change of primary emissions or meteorological influences, the implementation of the "Three-Year Action Plan for Blue Sky Defense" from 2018 to 2021 (www.gov.cn/zhengce/content/2018-07/03/content_5303158.htm) still resulted in a further decrease in eBC and then remained relatively stable at lower levels during 2019-2020."

There are a number of places that say ``similar to previous studies", or something like that. Can you also highlight what is new and different about your measurements that make your study unique?

The unique of this study is the long-term measurements and the more comprehensive analysis of eBC, BrC and aerosol optical properties, which provide important insights into the response of aerosol optical properties to clean air action, and the impact on radiative forcing. However, previous studies in Beijing either focused on BC, BrC or aerosol optical properties during specific periods, such long-term measurements and robust analysis have not been reported yet. The conclusions in previous studies that are important to justify our measurements and conclusions were cited in our manuscript.
Following the reviewer's suggestions, we expanded some discussions in the revised manuscript. For example,
 "The increase of MEE from 2.6 $m^2 g^{-1}$ in 2019 to 3.6 $m^2 g^{-1}$ in 2020 also suggested a significant change in scattering aerosol composition, such as an increase in nitrate contribution (Lei et al., 2020)".
 "The pronounced diurnal variations of SSA were characterized by afternoon peaks in all seasons, consistent with the measurements at 260 m in Beijing (Xie et al., 2019)".

Discuss Fig 10 some more. How significant are these numbers in terms of the total radiative balance? What implications for long-term changes to climate do these numbers show?

Following the reviewer's suggestions, we expanded the discussions in the revised manuscript.

"As shown in Fig. 8, the annual mean $\Delta F_R$ caused by BC was about +3.36 W m$^{-2}$ in 2012, close to that previously reported in north China (Yang et al., 2017). Combined with the low SSA (annual mean value was about 0.79) in 2012, the negative radiative effect caused by scatting aerosols at TOA might be offset by BC, forming an inversion layer that exacerbated air pollution. Ding et al. (2016) found that the aerosol-boundary layer feedback to unit quantity of BC will be lower in higher aerosol loading case as solar radiation weakened. However, $\Delta F_R$ decreased substantially by 67% (+1.09) in 2020, suggesting that the BC radiative forcing was largely reduced during last decade which would help improve the air quality by reducing aerosol-boundary interaction. The relatively lower $\Delta F_R$ caused by BC in recent years could lead to the negative radiative forcing of aerosols at TOA, thereby facilitate the dispersion of air pollutants in boundary layer, which will in turn maintain air pollution at a low level. However, the $\Delta F_R$ in 2020 was also much higher than the global annual mean TOA radiative forcing 0.40 W m$^{-2}$ (IPCC, 2013), indicating the positive radiative effect of BC in Beijing could be continually concerned in future."

The long-term measurements are essential to reduce the uncertainties of radiative forcing in climate models. The response of climate change to BC aerosol would be another interesting topic in the future study. For example, Zheng et al. (2020) evaluated the climate response to sustained changes in aerosol from 2006 to 2017 in China by using a climate model. The results showed that the reductions in aerosols in China may exert a mean warming of 0.12 °C in the Northern Hemisphere, and also affect the precipitation rates in East Asia. The success of Chinese policies to further reduce aerosol emissions may bring additional net warming.

Specific comments

Be precise about how you refer to ``9 years of data".    There are actually large gaps in your dataset, as presented in Fig S1.

Thanks for the comments. The time periods with the missing data were described in the manuscript. To clarify, we expanded more in the revised manuscript:

"Note that the measurements of Aethalometers and CAPS from June 2013 to September 2014 and from August 2015 to August 2017 were not available. The available data are from August 2012 to May 2013, October 2014 to September 2015 and September 2017 to December 2020." A more detailed instrument deployment is shown in Fig. S1.

Be consistent with how you refer to black carbon aerosol - you go back and forth between ``BC" and ``eBC".    I believe you are actually measuring eBC as you defined in the methods section.    Make sure to fix the abstract also.

We thank the reviewer for pointing this out. We have corrected the description of black carbon aerosol in the manuscript based on the principle as follows: eBC is used to mention our measurement of black carbon aerosol and BC is used in subtitle and

describing others' conclusion. We also clarified eBC in the abstract as equivalent BC (eBC).

A map of the study area showing the sampling locations would be nice and would provide context for Fig 5.
We thank for this suggestion and added the followed map figure in supplementary.

[Figure]

Fig. R4. Map of the sampling site and surrounding regions.

Are $b_{abs}$ and $b_{abs,BC}$ the same thing? I'm looking at equations 2 and 5.
We thank the reviewer for pointing this out. We have changed equations 2 as:

$$b_{abs,\ 630nm} = eBC \times MAE \tag{2}$$

and added the description as follows:
"After subtracting $b_{abs,\ BC}$ from the total absorption coefficient $b_{abs,\ total}$ at 370 nm, the BrC absorption coefficient ($b_{abs,\ BrC}$) can be estimated with Eq. (5)."

Lines 145, 148, 150 - why aren't these equations separated and numbered like others in the paper?
We have separated and numbered these equations as suggested in supplementary.

Lines 157 and 179 are essentially the same sentence.
We have removed the sentence in 179.

Fig 1 - the triangles are hard to see. Also, ``BC" or ``eBC"?

We thank the reviewer's suggestions and have made the following changes in Fig. 1.

[Figure]

Fig. R5. Annual variations of (a) eBC, $\triangle$eBC/$\triangle$CO, eBC/PM2.5, (b) $b_{ext}$, SSA and MEE. The median (horizontal line), mean (markers), 25th and 75th percentiles (lower and upper box), and 10th and 90th percentiles (lower and upper whiskers) are also shown, same as below.

Fig 2 - need error bars showing uncertainty or scatter in the data. Otherwise, we don't know how significant the monthly variations are.

We thank for the reviewer's comment and have added 25th and 75th percentiles in the figure, as shown below.

[Figure]

Fig. R6. Monthly variations in (a) eBC, (b) $\triangle$eBC/$\triangle$CO and (c) eBC/PM$_{2.5}$. The mean (markers), 25th and 75th percentiles (sticks) are also shown.

Line 175 - actually 2014 had an increase too

Because only two seasons of the eBC data in 2014 were available we did not make the conclusion about the increase in eBC in 2014. According to the review's comment, we have changed the description as "the mass concentrations of eBC were ubiquitously decreased in all seasons in 2020 compared to 2012."

Should ``clean air action plan'' be capitalized as a specific document or law?

We have added the site of clean air action plan as:

"However, our understanding of the long-term change of black carbon, aerosol optical properties and radiative effects as a response to the "Atmospheric Pollution Prevention and Control Action Plan" (www.gov.cn/zwgk/2013-09/12/content_2486773.htm) is very limited."

Line 215 - Why is 2012 an exception?

This is largely due to the low wind speeds in the summer of 2012, which made regional transport weaker. Thus, the pollutants are mainly from local emissions in 2012 which

seems different from other years.

Line 218 - Why is regional transport only important some of the time?
Previous studies have demonstrated the importance of regional pollution depending on meteorological conditions. For example, Guo et al. (2014) elucidated that urban haze is regulated by cyclical meteorological conditions. Chen et al. (2021) found that different contribution of regional transport to $PM_{2.5}$ occurred in different season based on WRF/CMAQ, e.g., local emissions dominate in winter and regional transport in summer. In particular, some severe polluted events are directly caused by meteorological variability in Beijing (Zhong et al., 2021). Therefore, different meteorological conditions associated with different sources around Beijing result in different relative importance (compared to local emissions) of regional transport to Beijing.

Line 315 - How do you know 2020 had a stronger atmospheric oxidation capacity?
We found that the ratio of BrC absorption to the total absorption increased to 16% from 2018 to 2020 with the significantly descending $b_{abs, Primary BrC}/b_{abs, Total BrC}$ ratios (75 % to 50 %). We then inferred these changes were likely due to enhanced photochemical production associated with stronger atmospheric oxidation capacity. We thank the reviewer for pointing this out, and we changed the statement as below:
Note that the average contribution of BrC to the total absorption in summer increased to 16 % from 2018 to 2020 likely due to enhanced secondary organic aerosol in OA (Lei et al., 2020), and correspondingly the contributions of primary BrC to the total BrC absorption decreased from 75 % to 50 %.

Figure 5 - Why not use the same color scale on each plot?
We have changed the color scale as suggested.

[Figure]

Fig. R7. Bivariate polar plots for hourly eBC mass concentration in the four seasons over nine years.

Is it necessary to show the top rows of Figures 6 and 7? These are just Figures 3 and 4 multiplied by a constant and therefore don't show anything new.

In our study, the directly measured data are $b_{ext}$ and light absorption at seven wavelengths, while the other variables were calculated based on the measured data. Therefore, we present both results in the manuscript so that the readers don't need to multiply the constants to get the values by themselves. We appreciate the suggestion and have merged Fig.3 with Fig.6, and Fig.4 with Fig.7 to simplify the illustration.

[Figure]

Fig. R8. Seasonal variations in eBC, eBC/PM$_{2.5}$, $\triangle$eBC/$\triangle$CO, CO, $b_{ext}$, SSA, PM$_{2.5}$ and MEE.

[Figure]

Fig. R9. Seasonal variations in eBC, eBC/PM$_{2.5}$, △eBC/△CO, CO, $b_{ext}$, SSA, PM$_{2.5}$ and MEE.

Fig 8 - the colors are hard to see
We have changed the colors in Fig.8 as below:

[Figure]

Fig. R10. Seasonal variations of (a) $b_{abs, BrC}$, AAE, percentage contribution of (b) absorbing components to the absorption coefficient and (c) the proportion of $b_{abs, Secondary BrC}$ in BrC absorption coefficient at 370nm from 2018 to 2020.

Technical comments

There are quite a few technical corrections to be made for grammar, word choice, etc. Only a few will be listed here; there are too many to write all of them out. One common mistake is missing articles such as ``the''. Another common confusion is the use of ``this'' and ``it'' as subjects in a sentence when these terms are no always clear as to what they are referring to.
Thanks for the reviewer's comments, we have corrected them as much as we can.

Line 244 - do you mean Fig 6?

Yes, we have corrected the figure number.

Line 269 - ``similar'' to what?
We thank the reviewer for pointing this out. It referred to "similar diurnal variation".

To clarify this, we revised this sentence as "Over the nine years, the diurnal variation of $b_{ext}$ was characterized by higher values at night and lower values during the day in all seasons."

Line 307 - ``similar" to what?
We revised this sentence as "As shown in Fig. S8, the monthly variation of BrC absorption was pronounced and similar to that of eBC, with high values in January and low values in July."

Line 351 - ``POA" is never defined
POA was defined as "primary organic aerosol" in the revised manuscript.

Check the bibliography carefully; there are formatting errors throughout.
Thanks for the reviewer's comment, we have checked the bibliography carefully in the revised manuscript.

**Response to Reviewer #2**

The manuscript by Sun et al. investigated black carbon and aerosol optical properties from 2012 to 2020 in Beijing, which is an important work to understand long-term changes of black carbon, aerosol optical properties and radiative effects after clean air action. Here, the authors applied a relatively mature experimental design to conduct a long-term observation. The results showed that large reductions in eBC and light extinction coefficient of fine particles were about 67 % and 47 %, respectively. And the most significant reductions of eBC were occurred in the fall and night time, mainly due to the changed primary emissions. Comparatively, SSA and MEE considerable increases highlight an increasing importance of scattering aerosols in radiative forcing, and a future challenge in visibility improvement due to enhanced MEE. Besides, the authors also quantified the primary and secondary BrC, demonstrating an enhanced role of secondary formation in BrC in recent years. In the last, the influences on direct radiative forcing from changes in BC and BrC are estimated. The whole result analysis in this study is systematical, thoughtful and in a well-organized manner. According to the state of the art, and the conclusions are supported by the data, so I consider that it fulfils the necessary requirements to be published. I recommend it for publication on Atmospheric Chemistry and Physics after the authors consider several minor revisions to the manuscript.

We thank the reviewer's positive comments.

Comments:

Please clarify eBC in abstract.

eBC was defined in the abstract as equivalent BC (eBC).

line 30&34, "during 2018-2020" format should be consistent, please check the full text.

We changed to a consistent format as "during 2018 – 2020".

line 63, "accounting for 25 ~ 35% 2010" should be "accounting for 25 ~ 35% in 2010".

We have changed as suggested.

line 84, ignore the punctuation. Please check the full text.

We added to the punctuation as: All optical measurements were conducted at the

Institute of Atmospheric Physics (IAP), Chinese Academy of Sciences (39°58′28″N,

116°22′16″E) in Beijing. More detail descriptions of this site were given in previous study. And check full text carefully.

line 107, Please check the Spaces after commas in the right subscript of the full text (e.g., babs, BC).

We changed a consistent format as $b_{abs, BC}$.

line 123, "the correlation (R2)" should be "the correlation coefficients (R2)".

Changed as suggested.

Line 160-170, it would be helpful to compare with a study about the seasonal variation of ΔBC/ΔCO in Beijing influenced by vertical transport of BC and wet scavenging. (Black carbon emission and wet scavenging from surface to the top of boundary layer over Beijing region, JGR-atmosphere, 125(17), 10.1029/2020JD033096, 2020.)

Added the reference as suggested.

line 163, Change "-" to "–"

Changed as suggested.

line 186 ~ 187, lack a preposition in this sentence. Please rewrite.

We rewrote this sentence as: Although eBC decreased by more than 60% in winter from 2012 to 2019, the different source contributions to BC were relatively constant in winter over eight years as indicated by flat $\triangle$eBC/$\triangle$CO (3 ~ 6 ng m$^{-3}$ ppbv$^{-1}$).

line 194, "the diurnal variation of eBC (Fig. 4) showed morning peaks" should be "the diurnal variations of eBC (Fig. 4) showed morning peaks".

Changed as suggested.

line 208, diurnal variation of winter in 2020 was not showed in Fig. 4. Please rewrite this sentence.

We rewrote the sentence as: " we found that the diurnal variations of both $\triangle$eBC/$\triangle$CO and eBC in 2019 and 2020 were less pronounced during four seasons."

line 209, "One reason was likely due to the fact that" could be "One reason was likely because"

Changed as suggested.

line 214, "the eBC presented similar distribution with high concentration in the middle and also the… "should be "the eBC presented similar distribution with high concentration in the middle and the…".

Changed as suggested.

line 242, "is close to" should be "was close to".

Changed as suggested.

line 244, "Fig. 7" should be "Fig. 6".

Corrected.

Please check the spaces of the full text (like line 107, e.g., babs, BC).

We thank the reviewer's comment and check full text carefully.

Line 310-315, a study which observed enhanced BrC contribution on absorption when heavy pollution could be referenced. (In situ vertical characteristics of optical properties and heating rates of aerosol over Beijing, Atmospheric Chemistry and Physics, 20(4), 2603–2622, 10.5194/acp-20-2603-2020, 2020.)

We thank the reviewer's suggestion and add the reference as suggested.

**Refrences**

Bond, T. C., Doherty, S. J., Fahey, D. W., Forster, P. M., Berntsen, T., DeAngelo, B. J., Flanner, M. G., Ghan, S., Kärcher, B., Koch, D., Kinne, S., Kondo, Y., Quinn, P. K., Sarofim, M. C., Schultz, M. G., Schulz, M., Venkataraman, C., Zhang, H., Zhang, S., Bellouin, N., Guttikunda, S. K., Hopke, P. K., Jacobson, M. Z., Kaiser, J. W., Klimont, Z., Lohmann, U., P., S. J., Shindell, D., Storelvmo, T., Warren, S. G., and Zender, C. S.: Bounding the role of black carbon in the climate system: A scientific assessment, J. Geophys. Res.-Atmos., 118, 5380-5552, 10.1002/jgrd.50171, 2013.

Chen, D., Xia, L., Guo, X., Lang, J., Zhou, Y., Wei, L., and Fu, X.: Impact of inter-annual meteorological variation from 2001 to 2015 on the contribution of regional transport to PM2.5 in Beijing, China, Atmos. Environ., 260, 118545, https://doi.org/10.1016/j.atmosenv.2021.118545, 2021.

[revised manuscript text omitted]

---

## Author Response (AR2)

**Response to Dr. Bahreini's comments**

We really appreciate the editor Roya Bahreini's comments on our manuscript which help improve the robustness of our conclusions. Following the suggestions, we have revised the manuscript accordingly. Listed below is our point-to-point response in blue to each comment.

1. With the MRS approach, the inherent assumption is that the sources of BC and BrC are different. However, in the real world, many of the processes that emit BC directly also emit BrC or its precursors, so in reality there is a correlation between BC and BrC (even if you assume BrC is only secondary). In this case, how valid is the interpretation of the MRS approach?

We agree with the editor that many processes emit both BC and BrC, for instance, biomass burning. The directly emitted BrC is defined as primary BrC in our study. The MRS approach is mainly used to differentiate the secondary BrC from primary BrC which has been reported in previous studies (Wang et al., 2019b; Zhu et al., 2021). Based on the investigation of MRS method in a previous study (Wu and Yu, 2016), the key of this method is to determine an appropriate $b_{abs,BrC}/eBC$ ratio that represents primary combustion emission sources, i.e., $(b_{abs,BrC}/eBC)_{pri}$ at the observation site. The hypothetical $(b_{abs,BrC}/eBC)_{pri}$ that generates a minimum $R^2$ (secondary BrC vs. eBC) represents the actual $(b_{abs,BrC}/eBC)_{pri}$ ratio if the variations of eBC and secondary BrC are independent and $(b_{abs,BrC}/eBC)_{pri}$ is relatively constant during the study period. Wu and Yu (2016) used numerically simulated data to evaluate the accuracy of estimated secondary components by the MRS method and also compared with the results from two commonly used methods. The bias of MRS result is <23% when the measurement uncertainty is within 20%. In addition, the MRS method can reduce the uncertainty caused by the different arbitrary selection criterion of $(b_{abs,BrC}/eBC)_{pri}$. Therefore, previous studies suggested that the MRS method is a useful method to distinguish primary and secondary components.

One assumption of the MRS method is that the sources of BC and secondary BrC are different although the sources of BC and primary BrC can be the same. This assumption might have some uncertainties in remote environment where BC and secondary aerosol are both from long-range transport. However, in urban atmosphere, secondary organic aerosol (SOA) that is mainly from the oxidation of volatile organic compounds (VOCs) (Hallquist et al., 2009) often exhibits largely different variations from primary organic aerosols, such as biomass burning OA, traffic-related OA, and coal combustion OA (Cappa et al., 2019; Xu et al., 2021). As part of the organic carbon, secondary BrC in this study is also expected to show different variations with primary components. To verify the results of secondary BrC absorption, we further calculated the secondary BrC absorption at 470 nm, 520 nm, 590 nm and 660 nm with the MRS approach, and then determined the AAE of secondary BrC. As shown in Figure R1, within the 10th to 90th percentiles, the AAE distribution of secondary BrC in our study ranged from 2 to 8, which is consistent with the AAE values of secondary organic carbon reported in

previous studies (Cheng et al., 2016; Qin et al., 2018; Kasthuriarachchi et al., 2020; Soleimanian et al., 2020; Kaskaoutis et al., 2021). Based on the above analysis, we concluded that the MRS approach is reasonable to determine the secondary BrC in urban area although there may have some uncertainties we cannot quantify now.

[Figure]

Figure R1. Annual variations of AAE of secondary BrC. The median (horizontal line), mean (markers), 25th and 75th percentiles (lower and upper box), and 10th and 90th percentiles (lower and upper whiskers) are also shown.

2. In the calculation of the radiative forcing values, constant back scatter fractions were assumed. However, the backscatter fraction is aerosol size and shape dependent. Please justify use of the constants here.

According to previous studies in Beijing, we found that BC exhibited an approximately lognormal size distribution with the median diameter about 170~210 nm, and the size distribution also showed some differences in different seasons (Liu et al., 2019; Yang et al., 2019; Liu et al., 2020b). Therefore, we did not analyze the variation of $\Delta F_R$ in different seasons in our study, instead, we focus on the changes of $\Delta F_R$ in the same season when the changes in size distribution of BC from 2013 to 2019 was small (Figure R2) (Wu et al., 2021), and the difference in annual median radius difference was less than 20 nm. Such a small difference affected the backscatter fraction of BC less than 0.04 (Chylek and Wong, 1995). Liu et al. (2017) found the difference of asymmetry factor (g) below 0.01 when the range of BC particle diameter was at 150~250 nm. Considering g as a function of backscatter fraction (Sviridenkov, 2015; Horvath et al., 2016), we further calculated the difference of BC particle backscattering fraction at 170 ~ 210 nm, and it was below 0.005. Therefore, the uncertainty of backscatter fraction caused by same change in size distribution of BC could be negligible in our study.

Besides, as the editor mentioned, the backscatter fraction also depends on BC particle shape. However, the effect of BC shape on the radiative effect is likely much lower than that caused by the large decrease (71%) in mass concentration. In previous studies, the

key parameter to represent the shape of BC particles is fractal dimension ($D_f$) (China et al., 2013). Previous studies found that most $D_f$ of BC particles from biomass burning and fossil fuel combustion was at 1.7~2.2 (Wei et al., 2020). According to the relationship between $D_f$, BC diameter and g in previous study (Liu et al., 2017), the maximum g difference of BC particles was at ~180 nm in the range of BC particle diameters of 150 ~ 250 nm (median mass diameter in recent years). At the same time, the range of g was only 0.47~0.45, corresponding to a $D_f$ of 1.8~2.8, respectively. We then calculated the backscattering fraction according to the relationship with g (Sviridenkov, 2015; Horvath et al., 2016) and found that the difference in the backscattering fraction was about 0.012.

Therefore, we considered these uncertainties to be acceptable and chose an appropriate backscatter fraction (Charlson et al., 1992; Chylek and Wong, 1995) which could fit the usual size distribution of BC in Beijing (Schwarz et al., 2008; Liu et al., 2019; Liu et al., 2020a; Liu et al., 2020b; Wu et al., 2021). To clarify this, we added the description about the backscatter fraction ($\beta$) as follows:

"Based on the small differences in mass size distribution of BC in recent years (Wu et al., 2021) and little effect of BC particle shape on backscatter fraction (Liu et al., 2017), we used a constant value of $\beta$ (0.29) (Charlson et al., 1992; Chylek and Wong, 1995) which could be adapted to the usual size distribution of BC in Beijing (Liu et al., 2019; Liu et al., 2020a; Liu et al., 2020b; Wu et al., 2021)."

[Figure]

**Fig. 1.** Mass size distributions of rBC ($dM/dlogD_c$) in urban Beijing in (a) winter, (b) spring, (c) summer, and (d) autumn normalized by the maximum $dM/dlogD_c$ values of respective distributions. The dashed gray lines represent the two modes combining the bimodal fitting.

Figure R2 (Wu et al., 2021).

3. Lastly, the SSA is also a function of aerosol size. With aerosol growth but under fixed composition, SSA increases. Therefore, a change in SSA alone cannot be taken as an indicator of a relative change between the contribution of the absorbing and non-absorbing species. Are there any measurements of aerosol size distributions that you can use to examine changes in SSA as a function of size parameter to be able to conclude any changes in the composition? If not, please elaborate on this caveat in the discussion.

We agree with the editor that aerosol optical properties are also related to particle size distributions in addition to chemical composition. As shown in Figure R3, the differences in aerosol volume size distribution from 2013 to 2019 in Beijing (available from the Aerosol Robotic Network data archive) are relatively small. Generally, the aerosol volume concentration exhibits a bimodal distribution and the first peak often occurs at ~200 nm except 2014 and 2019. These results indicate that the annual variation of aerosol optical properties (SSA and MEE) in recent years is more likely caused by the change in aerosol chemical components when the changes in aerosol particle size is relatively small.

[Figure]

Figure R3. Annual variations of aerosol volume size distribution in Beijing.

Previous studies also found that the aerosol volume size distributions showed seasonal variations in Beijing (Zheng et al., 2020). Following the editor's suggestion, we added an analysis of aerosol volume size distribution variation in different seasons (available from the Aerosol Robotic Network data archive). According to Figure R4, we found that the particle radius corresponding to the first volume concentration peak was the same at 200 nm in the fall of 2013 and 2018, while it became smaller in 2019. These results indicate that the increase in SSA in fall before 2019 was mainly due to the changes in aerosol chemical composition. Differently, the particle size in summer showed a yearly increasing trend from 2013 to 2018, and decreased in 2019 and

increased again in 2020. In spring in 2014 and 2015, higher volume concentrations occurred at larger radius (~180 nm) than those in other years. However, in winter the particle radius corresponding to the first volume concentration peak was larger in 2012 and 2018 (~200 nm) than other years. Because the measurements of particle size distributions were not available in this study, those derived from the Aerosol Robotic Network were mainly used to indicate the uncertainness in analyzing aerosol optical properties.

Following the editor's suggestion, we expanded the discussions of aerosol optical properties in the revised manuscript. In addition, we added the Figures R4 and R5 in supplementary to let readers know the uncertainties in this study.

The revised part in section 3.2 are as follows:
"In particular, the annual mean MEE and SSA increased despite the decreases in eBC and $b_{ext}$ and the relatively stable variations in particle size distribution (Fig. S7) in the past decade, which indicated that scattering aerosol species played more important roles than absorbing aerosol species in radiative forcing."

"While the increase of SSA in fall was mainly due to the changes in aerosol chemical composition such as the decrease in BC, the increase of SSA in summer was also likely affected by the increase in aerosol particle sizes. These results were supported by the similar particle size (~ 200 nm) in fall, while a clear increase in summer from 2013 to 2018 (Fig.S8). Note that the particle size in summer 2019 was smaller than that in 2018, yet the seasonal mean SSA was higher in 2019, suggesting that the changes in aerosol chemical components contributed significantly to the increased SSA."

"Considering the increased SSA and particle size (Fig. S8) yet relatively constant mass concentrations of eBC during 2017 ~2019, we inferred that the increased light extinction in winter was mainly caused by scattering aerosols that can vary substantially in different years due to the changes in meteorological conditions (Zhou et al., 2019)."

"The diurnal cycles of $b_{ext}$ and SSA in four seasons are shown in Fig. 4. Because the diurnal variations of particle size distributions are not available in this study, our discussions are mainly focused on the influences of chemical components."

[Figure]

Figure R4. Seasonal variations of aerosol volume size distribution in Beijing.

**References**

Cappa, C. D., Zhang, X., Russell, L. M., Collier, S., Lee, A. K. Y., Chen, C.-L., Betha, R., Chen, S., Liu, J., Price, D. J., Sanchez, K. J., McMeeking, G. R., Williams, L. R., Onasch, T. B., Worsnop, D. R., Abbatt, J., and Zhang, Q.: Light absorption by ambient black and brown carbon and its dependence on black carbon coating state for two California, USA, cities in winter and summer, J. Geophys. Res.-Atmos., 124, 1550-1577, 10.1029/2018jd029501, 2019.

Charlson, R. J., Schwartz, S. E., Hales, J. M., Cess, R. D., Coakley, J. A., Hansen, J. E., and Hofmann, D. J.: Climate forcing by anthropogenic aerosols, Science, 255, 423-430, 1992.

Cheng, Y., He, K.-b., Du, Z.-y., Engling, G., Liu, J.-m., Ma, Y.-l., Zheng, M., and Weber, R. J.: The characteristics of brown carbon aerosol during winter in Beijing, Atmos. Environ., 127, 355-364, 10.1016/j.atmosenv.2015.12.035, 2016.

China, S., Mazzoleni, C., Gorkowski, K., Aiken, A. C., and Dubey, M. K.: Morphology and mixing state of individual freshly emitted wildfire carbonaceous particles, Nat. Commun., 4, 2122, 10.1038/ncomms3122, 2013.

Chylek, P. and Wong, J.: Effect of aerosols on global budget, Geophys. Res. Lett., 22, 929-931, 1995.

Hallquist, M., Wenger, J. C., Baltensperger, U., Rudich, Y., Simpson, D., Claeys, M., Dommen, J., and Donahue, N. M., George, C., Goldstein, A. H., Hamilton, J. F., Herrmann, H., Hoffmann, T., Iinuma, Y., Jang, M., Jenkin, M. E., Jimenez, J. L., Kiendler-Scharr, A., Maenhaut, W., McFiggans, G., Mentel, Th. F., Monod, A., Prévôt, A. S. H., Seinfeld, J. H., Surratt, J. D., Szmigielski, R., and Wildt, J.: The formation, properties and impact of secondary organic aerosol: current and emerging issues, Atmos. Chem. Phys. Discuss, 9, 5155–5236, 2009.

Horvath, H., Kasahara, M., Tohno, S., Olmo, F. J., Lyamani, H., Alados-Arboledas, L., Quirantes, A., and Cachorro, V.: Relationship between fraction of backscattered light and asymmetry parameter, J. Aerosol. Sci., 91, 43-53, 10.1016/j.jaerosci.2015.09.003, 2016.

Hu, W., Hu, M., Hu, W.-W., Zheng, J., Chen, C., Wu, Y., and Guo, S.: Seasonal variations in high time-resolved chemical compositions, sources, and evolution of atmospheric submicron aerosols in the megacity Beijing, Atmos. Chem. Phys., 17, 9979-10000, 10.5194/acp-17-9979-2017, 2017.

Kaskaoutis, D. G., Grivas, G., Stavroulas, I., Bougiatioti, A., Liakakou, E., Dumka, U. C., Gerasopoulos, E., and Mihalopoulos, N.: Apportionment of black and brown carbon spectral absorption sources in the urban environment of Athens, Greece, during winter, Sci. Total Environ., 801, 149739, 10.1016/j.scitotenv.2021.149739, 2021.

Kasthuriarachchi, N. Y., Rivellini, L.-H., Adam, M. G., and Lee, A. K. Y.: Light absorbing properties of primary and secondary brown carbon in a tropical urban environment, Environ. Sci. Technol., 54, 10808-10819, 10.1021/acs.est.0c02414, 2020.

Liu, C., Li, J., Yin, Y., Zhu, B., and Feng, Q.: Optical properties of black carbon aggregates with non-absorptive coating, J. Quant. Spectrosc. Radiat. Transfer, 187, 443-452, 10.1016/j.jqsrt.2016.10.023, 2017.

Liu, D., Joshi, R., Wang, J., Yu, C., Allan, J. D., Coe, H., Flynn, M. J., Xie, C., Lee, J.,

Squires, F., Kotthaus, S., Grimmond, S., Ge, X., Sun, Y., and Fu, P.: Contrasting physical properties of black carbon in urban Beijing between winter and summer, Atmos. Chem. Phys., 19, 6749-6769, 10.5194/acp-19-6749-2019, 2019.

Liu, D., Ding, S., Zhao, D., Hu, K., Yu, C., Hu, D., Wu, Y., Zhou, C., Tian, P., Liu, Q., Wu, Y., Zhang, J., Kong, S., Huang, M., and Ding, D.: Black carbon emission and wet scavenging from surface to the top of boundary layer over Beijing region, J. Geophys. Res.-Atmos., 125, 10.1029/2020jd033096, 2020a.

Liu, H., Pan, X., Wu, Y., Ji, D., Tian, Y., Chen, X., and Wang, Z.: Size-resolved mixing state and optical properties of black carbon at an urban site in Beijing, Sci. Total Environ., 749, 141523, 10.1016/j.scitotenv.2020.141523, 2020b.

Qin, Y. M., Tan, H. B., Li, Y. J., Li, Z. J., Schurman, M. I., Liu, L., Wu, C., and Chan, C. K.: Chemical characteristics of brown carbon in atmospheric particles at a suburban site near Guangzhou, China, Atmos. Chem. Phys., 18, 16409-16418, 10.5194/acp-18-16409-2018, 2018.

Schwarz, J. P., Gao, R. S., Spackman, J. R., Watts, L. A., Thomson, D. S., Fahey, D. W., Ryerson, T. B., Peischl, J., Holloway, J. S., Trainer, M., Frost, G. J., Baynard, T., Lack, D. A., de Gouw, J. A., Warneke, C., and Del Negro, L. A.: Measurement of the mixing state, mass, and optical size of individual black carbon particles in urban and biomass burning emissions, Geophys. Res. Lett., 35, 10.1029/2008gl033968, 2008.

Shrivastava, M., Cappa, C. D., Fan, J., Goldstein, A. H., Guenther, A. B., Jimenez, J. L., Kuang, C., Laskin, A., Martin, S. T., Ng, N. L., Petaja, T., Pierce, J. R., Rasch, P. J., Roldin, P., Seinfeld, J. H., Shilling, J., Smith, J. N., Thornton, J. A., Volkamer, R., Wang, J., Worsnop, D. R., Zaveri, R. A., Zelenyuk, A., and Zhang, Q.: Recent advances in understanding secondary organic aerosol: Implications for global climate forcing, Rev. Geophy., 55, 509-559, 10.1002/2016rg000540, 2017.

Soleimanian, E., Mousavi, A., Taghvaee, S., Shafer, M. M., and Sioutas, C.: Impact of secondary and primary particulate matter (PM) sources on the enhanced light absorption by brown carbon (BrC) particles in central Los Angeles, Sci. Total Environ., 705, 135902, 10.1016/j.scitotenv.2019.135902, 2020.

Sun, Y., Xu, W., Zhang, Q., Jiang, Q., Canonaco, F., Prévôt, A. S. H., Fu, P., Li, J., Jayne, J., Worsnop, D. R., and Wang, Z.: Source apportionment of organic aerosol from 2-year highly time-resolved measurements by an aerosol chemical speciation monitor in Beijing, China, Atmos. Chem. Phys., 18, 8469-8489, 10.5194/acp-18-8469-2018, 2018.

Sviridenkov, M. A.: Retrieval of asymmetry factor of scattering phase function from scattering and extinction measurements, Presented at the European Aerosol Conference, Milan, Italy, September 6 to 11, 2015., 2015.

Wang, Q., Han, Y., Ye, J., Liu, S., Pongpiachan, S., Zhang, N., Han, Y., Tian, J., Wu, C., Long, X., Zhang, Q., Zhang, W., Zhao, Z., and Cao, J.: High Contribution of Secondary Brown Carbon to Aerosol Light Absorption in the Southeastern Margin of Tibetan Plateau, Geophys. Res. Lett., 46, 4962-4970, 10.1029/2019gl082731, 2019a.

Wang, Q., Ye, J., Wang, Y., Zhang, T., Ran, W., Wu, Y., Tian, J., Li, L., Zhou, Y., Hang Ho, S. S., Dang, B., Zhang, Q., Zhang, R., Chen, Y., Zhu, C., and Cao, J.: Wintertime optical properties of primary and secondary brown carbon at a regional site in the North China Plain, Environ. Sci. Technol., 53, 12389-12397, 10.1021/acs.est.9b03406, 2019b.

Wei, X., Zhu, Y., Hu, J., Liu, C., Ge, X., Guo, S., Liu, D., Liao, H., and Wang, H.: Recent Progress in Impacts of Mixing State on Optical Properties of Black Carbon Aerosol, Current Pollution Reports, 6, 380-398, 10.1007/s40726-020-00158-0, 2020.

Wu, C. and Yu, J. Z.: Determination of primary combustion source organic carbon-to-elemental carbon (OC / EC) ratio using ambient OC and EC measurements: secondary OC-EC correlation minimization method, Atmos. Chem. Phys., 16, 5453-5465, 10.5194/acp-16-5453-2016, 2016.

Wu, Y., Xia, Y., Zhou, C., Tian, P., Tao, J., Huang, R. J., Liu, D., Wang, X., Xia, X., Han, Z., and Zhang, R.: Effect of source variation on the size and mixing state of black carbon aerosol in urban Beijing from 2013 to 2019: Implication on light absorption, Environ. Pollut., 270, 116089, 10.1016/j.envpol.2020.116089, 2021.

Xu, W., Chen, C., Qiu, Y., Li, Y., Zhang, Z., Karnezi, E., Pandis, S. N., Xie, C., Li, Z., Sun, J., Ma, N., Xu, W., Fu, P., Wang, Z., Zhu, J., Worsnop, D. R., Ng, N. L., and Sun, Y.: Organic aerosol volatility and viscosity in the North China Plain: contrast between summer and winter, Atmos. Chem. Phys., 21, 5463-5476, 10.5194/acp-21-5463-2021, 2021.

Yang, Y., Xu, X., Zhang, Y., Zheng, S., Wang, L., Liu, D., Gustave, W., Jiang, L., Hua, Y., Du, S., and Tang, L.: Seasonal size distribution and mixing state of black carbon aerosols in a polluted urban environment of the Yangtze River Delta region, China, Sci. Total Environ., 654, 300-310, 10.1016/j.scitotenv.2018.11.087, 2019.

Zheng, Y., Che, H., Xia, X., Wang, Y., Yang, L., Chen, J., Wang, H., Zhao, H., Li, L., Zhang, L., Gui, K., Yang, X., Liang, Y., and Zhang, X.: Aerosol optical properties and its type classification based on multiyear joint observation campaign in north China plain megalopolis, Chemosphere, 128560, 10.1016/j.chemosphere.2020.128560, 2020.

Zhou, W., Xu, W., Kim, H., Zhang, q., Fu, P., Worsnop, D., and Sun, Y.: A review of aerosol chemistry in Asia: insights from aerosol mass spectrometer measurements, Environmental Science: Processes & Impacts, 10.1039/D0EM00212G, 2020.

Zhu, C.-S., Qu, Y., Zhou, Y., Huang, H., Liu, H.-K., Yang, L., Wang, Q.-Y., Hansen, A. D. A., and Cao, J.-J.: High light absorption and radiative forcing contributions of primary brown carbon and black carbon to urban aerosol, Gondwana Research, 90, 159-164, 10.1016/j.gr.2020.10.016, 2021.

---

## Author Response (AR3)

Response to Dr. Bahreini's comments

We appreciate the editor Roya Bahreini's further comments on our manuscript. Following her comments, we revised the manuscript again. Listed below is our point-to-point response in blue to each comment.

- change "stable" to "minor".
Changed.

- change line 241 to read as "These results were supported by the similar particle size distributions (mode ~ 200 nm) in fall, while a clear increase in the mode was observed during summer from 2013 to 2018 (Fig.S8)."
Changed as suggested.

- change the units of volume distribution plots to say um3/cm3 or something similar (it's now volume per area).
We thank Dr. Bahreini's comment on the unit of volume distributions. Because the volume size distribution is retrieved from spectral optical thickness and represents the columnar volume size distribution, thus the unit is $\mu m^3/\mu m^2$, which is different from that measured by SMPS, i.e., $\mu m^3/cm^3$. We therefore kept the unit in the supplementary.

- change "scatting" to "scattering" in line 252.
Changed.